# PKCε switches Aurora B specificity to exit the abscission checkpoint

Tanya Pike[1], Nicola Brownlow[1,†], Svend Kjaer[2], Jeremy Carlton[3] & Peter J. Parker[1,3]

The 'NoCut', or Aurora B abscission checkpoint can be activated if DNA is retained in the cleavage furrow after completion of anaphase. Checkpoint failure leads to incomplete abscission and a binucleate outcome. These phenotypes are also observed after loss of PKCε in transformed cell models. Here we show that PKCε directly modulates the Aurora B-dependent abscission checkpoint by phosphorylating Aurora B at S227. This phosphorylation invokes a switch in Aurora B specificity, with increased phosphorylation of a subset of target substrates, including the CPC subunit Borealin. This switch is essential for abscission checkpoint exit. Preventing the phosphorylation of Borealin leads to abscission failure, as does expression of a non-phosphorylatable Aurora B S227A mutant. Further, depletion of the ESCRT-III component and Aurora B substrate CHMP4C enables abscission, bypassing the PKCε–Aurora B exit pathway. Thus, we demonstrate that PKCε signals through Aurora B to exit the abscission checkpoint and complete cell division.

[1] Protein Phosphorylation Laboratory, The Francis Crick Institute, 1 Midland Road, London NW1 1AT, UK. [2] Protein Purification Facility, Francis Crick Institute, 1 Midland Road, London NW1 1AT, UK. [3] Division of Cancer Studies King's College London, New Hunt's House, Guy's Campus, London SE1 1UL, UK. † Present address: Centre for Genomic Regulation (CRG), Dr Aiguader, 88, PRBB Building, 08003 Barcelona, Spain. Correspondence and requests for materials should be addressed to P.J.P. (email: peter.parker@crick.ac.uk).

The existence of a final 'NoCut' checkpoint before exit from cytokinesis has been defined by a number of groups in yeast, worms and mammalian systems[1–7]. This checkpoint, operating at the point of no return for precise and successful self-renewal, is dependent on Aurora B kinase activity and is engaged by the presence of chromatin trapped in the cytokinesis furrow. The localization of Aurora B to key mitotic structures throughout mitosis, including the midbody during cytokinesis indicates a fundamental role for the kinase in successful completion of mitosis[8]. Aurora B and components of the chromosomal passenger complex (CPC) are known to interact with and trigger phosphorylation of downstream substrates to effect completion of cytokinesis; these substrates include the following: MgcRacGap[9]; vimentin[10]; PRC1 (ref. 11); MKLP1 (ref. 12); and CHMP4C (refs 5,13).

The final stage of cytokinesis, abscission, follows ordered recruitment of the endosomal sorting complexes required for transport (ESCRT) machinery to the midbody leading to constriction of ESCRT-III filaments bringing the membranes together until scission can occur (reviewed in ref. 14). The ESCRT-III subunit CHMP4C is a key player in the Aurora B abscission checkpoint, stalling abscission due to the presence of retained chromatin in the furrow through interaction with the CPC subunit Borealin and phosphorylation by Aurora B[5,15]. Phosphorylation of CHMP4C at S210 maintains the subunit in a closed, inactive conformer, unable to polymerize with other ESCRT-III subunits[13]. The AAA-ATPase VPS4 has also been shown to be controlled in an Aurora B-dependent manner through CHMP4C association; VPS4 localization to the midbody ring is inhibited when the abscission checkpoint is engaged, thereby preventing abscission[4]. The tight regulation of this final process in cell division is the ultimate protective measure against a chromosome non-disjunction error that could lead to unequal inheritance between the two daughter cells and genome instability.

Protein kinase C (PKC) has been implicated in completion of the final stages of mitosis[16–19]. In particular, activity of the epsilon isoform (PKCε) has been shown to be required for successful exit from cytokinesis through a combination of signalling pathways involving RhoA (refs 17,18) and ZO-1 (ref. 19). Knockdown or inhibition of PKCε results in failure to complete abscission, the final step in severing the bridge between the two daughter cells; abscission failure typically leads to polyploidization. PKCε, unlike Aurora B, has a unique role in these control processes not being required in 'normal' diploid cells but seemingly playing a critical role in a subset of transformed cells[18,20]. This prompted us to investigate whether this PKCε conditional action was linked to the engagement of Aurora B in the abscission pathway.

Here we report PKCε regulation of Aurora B is required for exit from the abscission checkpoint. We demonstrate that PKCε directly phosphorylates Aurora B at S227 when localized to the midbody. S227 phosphorylation induces a switch in Aurora B substrate specificity promoting the phosphorylation of the CPC subunit Borealin at S165 to facilitate abscission exit. This signalling cascade results in the localization of the ESCRT-III component CHMP4C to the midbody ring enabling the final scission event.

## Results

### PKCε regulates Aurora B at the midbody during cytokinesis.
On inhibition, PKCε accumulates at and stabilizes the midbody[17], hence to assess what process(es) it might engage there, we screened a midbody protein-biased peptide array for candidate substrates of PKCε (Fig. 1a). This identified a number of candidates for which phosphorylation was greater than the optimized pseudosubstrate site peptide control (Supplementary Fig. 1a and Supplementary Data 1). Included in these 'hits' was the protein kinase Aurora B, where S227, a site proximal to the activation loop T232 phosphorylation site, was identified. An in vitro kinase assay with the two proteins demonstrated a twofold increase in $^{32}$P-phosphate incorporation into Aurora B catalysed by PKCε (Supplementary Fig. 1b). We raised an antibody selective for phosphorylated S227 in Aurora B, and this revealed PKCε phosphorylation of this Aurora B site in vitro (Fig. 1b). Phosphorylation of the S227 site has been identified in vivo in mass spectrometry screens (Supplementary Data 1), and notably the site is conserved within the Aurora family of kinases and indeed within higher eukaryotes (Fig. 1c).

On the basis of localization of the endogenous proteins by immunofluorescence (Supplementary Figs 1c and 2b) and more directly on proximity ligation assays, Aurora B and PKCε were found to co-localize at the midbody in DLD1 cells as observed for Aurora B and INCENP subunits of the CPC (Fig. 1d and Supplementary Fig. 1d). Furthermore, assessment of S227 phosphorylation in DLD1 cells demonstrated that a subset of Aurora B phosphorylated at this site localizes at the midbody (note the nuclear recognition is nonspecific and not eliminated by knockdown of Aurora B using siRNA (Supplementary Fig. 1e) nor expression of a non-phosphorylatable S227A mutant (Supplementary Fig. 1f; see further below)). The midbody phospho-S227 immunoreactivity is sensitive to inhibition of PKCε after a 30 min treatment with the selective inhibitor BLU577 (compound 18, ref. 21; Fig. 1e and Supplementary Fig. 1f). Analysis of S227 phosphorylation through mitosis revealed chromatin-associated staining of Aurora B S227 phosphorylation in mitosis (which was absent following Aurora B knockdown (Supplementary Fig. 1f, lower panel)) but not during telophase (Supplementary Fig. 1g). Phosphorylation of the activating T232 residue was not influenced by PKCε inhibition (Supplementary Fig. 1h), nor by expression of the S227A mutant of Aurora B (Supplementary Fig. 1i).

### Aurora B localization at the midbody.
On close inspection of the midbody, we observed that Aurora B phosphorylated on T232 localizes to the midbody arms, the midbody ring and overlays with total Aurora B staining albeit concentrated in the midbody ring (Supplementary Fig. 2a). By contrast Aurora B phosphorylated on S227 exclusively localizes to the midbody ring, We note here that Hu et al.[22] describe Aurora B (both endogenous and green fluorescent protein (GFP)-tagged) as being excluded from the stem body and localizing to the flanking regions, however we clearly observe this ring structure localization in both endogenous, antibody-stained samples (Supplementary Fig. 2a) and GFP-Aurora B-expressing cells (Supplementary Fig. 2b). Certainly, most of the Aurora B appears located on the midbody arms and it may be the case that it is difficult to visualize Aurora B that is localized in the midbody ring without specific antibodies for that compartment.

### Successful abscission requires Aurora B S227 phosphorylation.
To assess the role of this phosphorylation in Aurora B action, we established cell lines that express an inducible Aurora B wild type (WT) or the S227A mutant at similar levels (Supplementary Fig. 2c). Interestingly, expression of the S227A mutant increased the number of binucleated cells (Fig. 2a) as a result of abscission failure (Fig. 2b and Supplementary Fig. 2d), phenocopying PKCε inhibition. Cells expressing Aurora B S227A did not fail abscission due to a prolonged delay or arrest during cytokinesis (Supplementary Fig. 2e–g). Comparing Aurora B WT and S227A expressing cells, no significant increase in the number of cells in

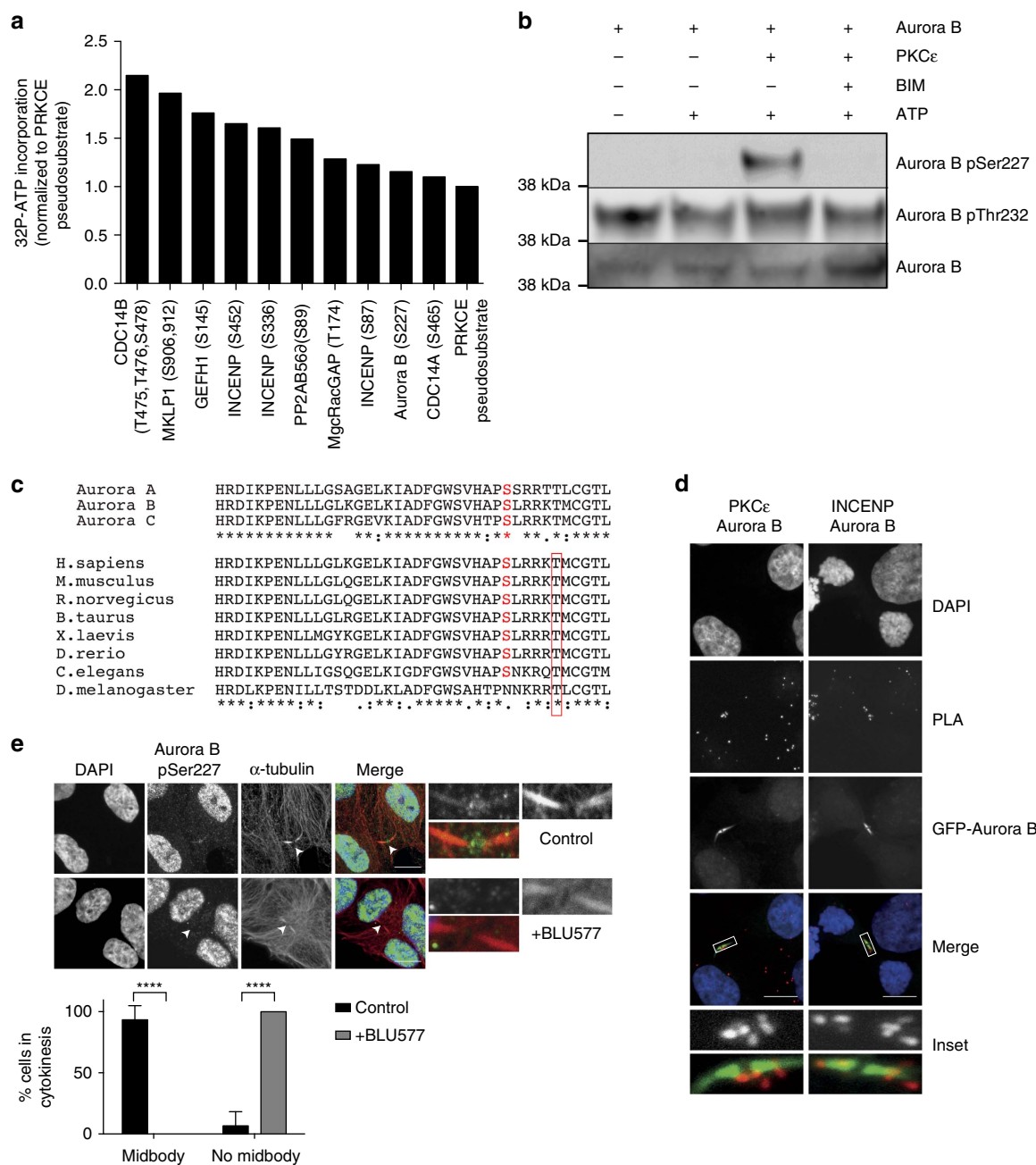

**Figure 1 | PKCε specifically phosphorylates Aurora B S227 at the midbody during cytokinesis.** (**a**) A midbody protein-biased peptide array of PKCε putative substrates was assayed for $^{32}$P-ATP incorporation and normalized to the pseudosubstrate peptide as a positive control. Data for all peptides analysed in Supplementary Data 1. (**b**) *In vitro* kinase assay of PKCε and Aurora B with and without the PKC inhibitor bis-indolylmeleimide I (BIM) (1 μm). (**c**) Protein sequence alignment of human Aurora kinase family members and Aurora B of various species. S227 (human Aurora B sequence) is highlighted in red, T232 is boxed in red. (**d**) Proximity ligation assay between Aurora B and PKCε in DLD1 GFP-Aurora B WT cell line. The interaction between CPC members Aurora B and INCENP served as a positive control for this assay. Primary antibodies against the endogenous proteins were used for the detection of Aurora B, PKCε and INCENP. DAPI (blue), proximity ligation assay (PLA) (red), GFP-Aurora B (green). Scale bar, 10 μm. (**e**) Confocal imaging shows Aurora B pS227 (green) staining at the cytokinesis midbody (tubulin—red) in DLD1 cells (white arrows and inset) and not in the presence of the PKCε selective inhibitor BLU577 (500 nM) for 30 min. Cells were scored for the presence or absence of Aurora B pS227 at the midbody (statistics analysed by Student's *t*-test; ****$P < 0.0001$; error bars represent mean ± s.e.m.). A minimum of 30 high-resolution, single-cell images per condition from 12 experiments in two different cell lines were acquired, a representative image is shown here. Scale bar, 10 μm.

cytokinesis (Supplementary Fig. 2e) nor difference in the time from telophase onset to midbody dissolution (for either successful or failed cytokinesis) was observed (Supplementary Fig. 2f,g). However, the behaviour of S227A-expressing cells was associated with an increase in DNA in the furrow as evidenced by a substantial increase in Lap2β-positive structures in the furrow

(Fig. 2c,d); again this phenocopied PKCε inhibition with BLU577. The abscission failure was not a function of mislocalization of Aurora B itself (Supplementary Fig. 2h) nor of any component of the CPC (Supplementary Fig. 3), as inhibition of PKCε with BLU577 or expression of the S227A mutant followed the same cell cycle pattern of subcellular distribution as the WT.

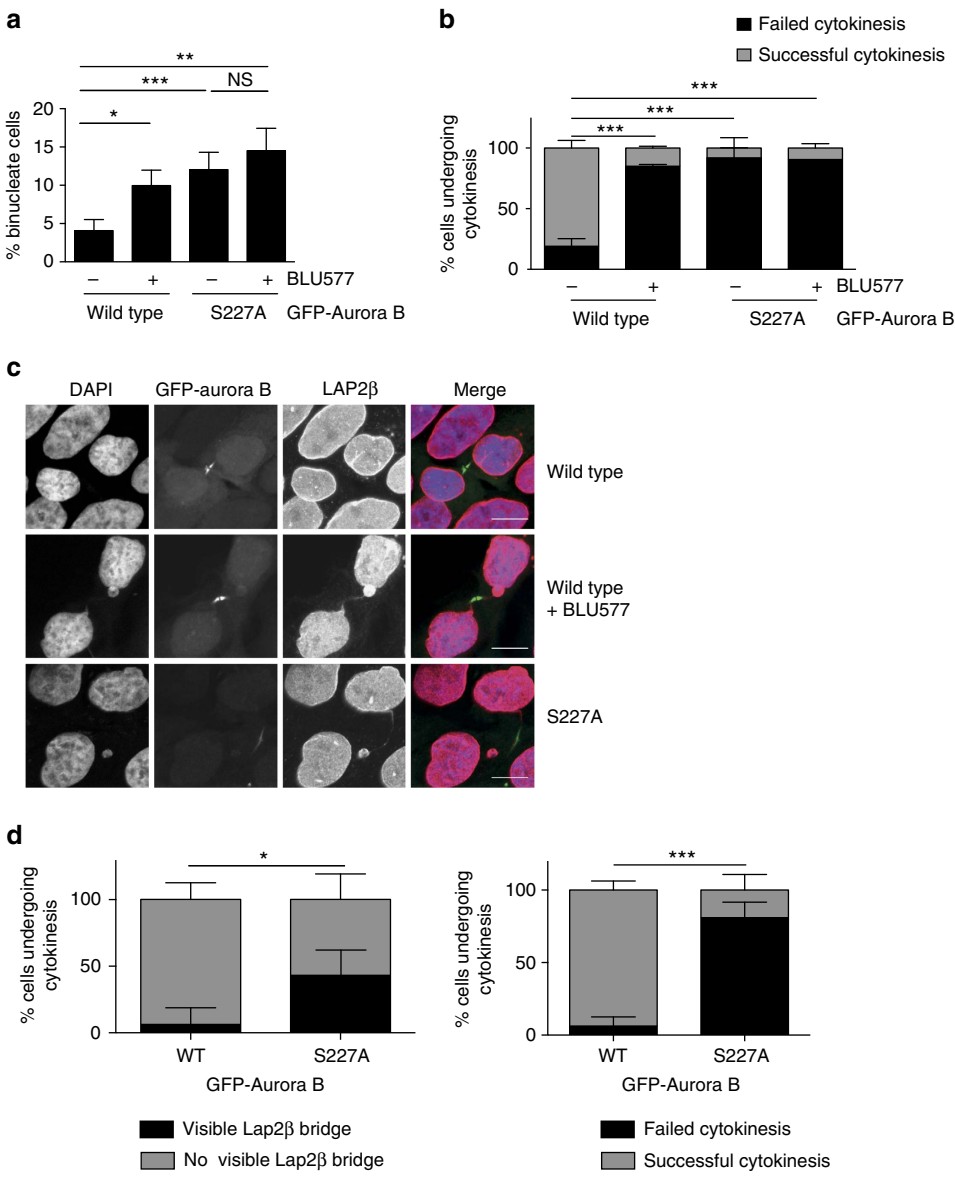

**Figure 2 | Aurora B S227 phosphorylation is required for successful completion of cytokinesis. (a,b)** Cells where PKCε is inhibited and/or Aurora B cannot be S227 phosphorylated do not undergo successful abscission. (**a**) The number of binucleate cells was assessed after 24 h induction of GFP-Aurora B WT (4.07% ± 1.45), GFP-Aurora B S227A (12.03% ± 2.27) ± BLU577 (500 nM) (WT 9.97% ± 2.01 versus S227A 14.5% ± 2.94) expression in the DLD1 cell lines. Graph represents the mean (± s.e.m.) of three independent experiments of >500 scored cells per condition. Student's *t*-test, not significant (NS) = $P > 0.05$, * = $P ≤ 0.05$, ** = $P ≤ 0.01$, *** = $P ≤ 0.001$. (**b**) Wide-field time-lapse microscopy of DLD1 GFP-Aurora B cell lines. Cells were scored for the outcome of cytokinesis where failed cytokinesis is a binucleate cell and successful cytokinesis resulted in two daughter cells. Cells were induced for GFP-Aurora B expression for 16 h before image acquisition. Graph represents the mean (± s.e.m.) of three independent experiments where a minimum of 100 cells were scored per condition. Two-way analysis of variance (ANOVA), *** = $P ≤ 0.001$. (**c,d**) There is an increase in cells with DNA trapped in the cytokinesis furrow if PKCε is inhibited and Aurora B cannot be phosphorylated on S227. (**c**) Confocal images of DLD1 GFP-Aurora B (green) cell lines stained for Lap2β (red) to identify DNA (DAPI—blue) bridging during cytokinesis. Scale bar, 10 μm. (**d**) DLD1 GFP-Aurora B cell lines, which stably express RFP-Lap2β were assessed for the presence of a Lap2β-positive bridge as they underwent cytokinesis (WT 6.25% ± 12.5 versus S227A 43.125% ± 19.2; left panel) and the outcome of cytokinesis (binucleate cells: WT 6.25% ± 12.5 versus S227A 80.9% ± 21.5; right panel). Graph represents the mean (± s.e.m.) of three independent experiments where a minimum of 100 cells scored per condition. Two-way ANOVA, * = $P ≤ 0.05$, *** = $P ≤ 0.001$.

**Phosphorylation of S227 alters Aurora B substrate repertoire.** It would be plausible to think that PKCε-mediated phosphorylation of Aurora B might be inhibitory to Aurora B, and this might trigger exit from the abscission checkpoint. However, comparing recombinant Aurora B that is S227/T232-phosphorylated (WT) or T232-phosphorylated only (S227A) demonstrated equivalent ATP-binding capabilities (Supplementary Fig. 4a) and also equal phosphotransferase activities towards the substrate histone H3 S10 (Supplementary Fig. 4b), indicating that

intrinsic catalytic activity is not affected. This led to the hypothesis that there was not a loss of function but a change of function associated with S227 phosphorylation, reflected in altered substrate recognition. To assess this hypothesis, we screened a peptide array of Aurora B substrate sites (Fig. 3a, Supplementary Fig. 4c and Supplementary Data 2). This revealed that compared with the S227A mutant, the doubly phosphory-lated protein has a strong preference and indeed much greater activity towards a subset of sites, with a >10-fold increase in

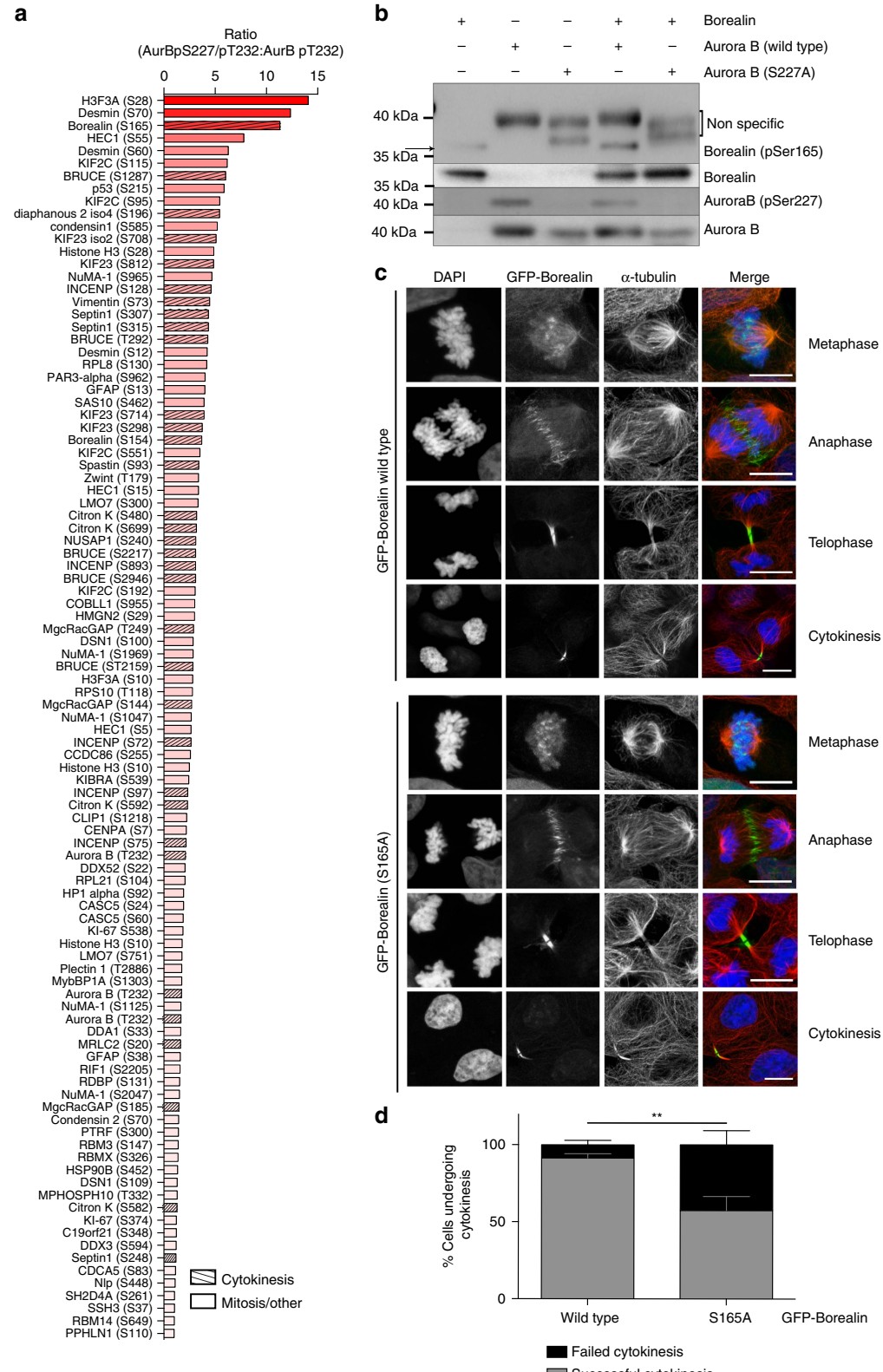

**Figure 3 | Aurora B phosphorylated on S227/T232 has unique set of substrates including Borealin S165.** (**a**) A peptide array of known Aurora B substrates compared the phosphorylation of peptides by either doubly phosphorylated (WT) Aurora B to those phosphorylated by singly phosphorylated (S227A) Aurora B. Graph shows ratio of ATP incorporation into peptides phosphorylated by Aurora B WT and S227A recombinant protein. Data for all peptides analysed in Supplementary Data 2. (**b**) *In vitro* kinase assay to detect Borealin Ser165 phosphorylation by recombinant Aurora B(WT) or (S227A) protein. (**c,d**) DLD1 cells were induced to express GFP-Borealin WT or GFP-Borealin S165A for 16 h before analysis. (**c**) Representative confocal images of GFP-Borealin cells in each phase of mitosis and cytokinesis. Scale bar, 10 μm. (**d**) DLD1 cells induced to express GFP-Borealin were scored for the outcome of cytokinesis, successful cytokinesis resulting in two daughter cells or failed cytokinesis resulting in a binucleate cell. Graph is the mean ( ± s.e.m.) of three independent experiments where more than 100 cells per condition were scored. Two-way analysis of variance, ** = $P \leq 0.01$.

phosphorylation of the three top hits: H3F3A; Desmin; and Borealin. Phosphorylation of Borealin S165 has been identified previously in a mitotic phosphoproteome screen[23] and was also recognized as a direct substrate of Aurora B[24,25]. As part of the chromosome passenger complex required for cytokinesis[25] and known to bind to abscission complex component CHMP4C (refs 5,13) this last target provided a candidate for relaying the effect of PKCε phosphorylation of Aurora B on S227.

A phospho-specific antibody was raised against Borealin S165, and by in vitro kinase assay with recombinant Aurora B we were able to detect phosphorylation at this site in the presence of WT (pS227/pT232) Aurora B but not Aurora B 227A (pT232 only) (Fig. 3b), confirming this site in vitro as a substrate for the S227/T232-phosphorylated Aurora B.

To determine whether Borealin phosphorylation was part of the regulatory cascade involved in this pathway, we stably expressed an inducible WT or S165A mutant GFP-Borealin in DLD1 cells (Fig. 3c). We observed that like the Aurora B S227A mutant, expression of the Borealin S165A mutant caused abscission failure in 42.7% (±14.7) cells undergoing cytokinesis (Fig. 3d). The implication is that with retained DNA in the furrow causing engagement of the Aurora B-dependent abscission checkpoint, PKCε acts to phosphorylate Aurora B, which in turn phosphorylates Borealin to trigger completion of abscission—failure of any of these steps causing abscission failure.

The ESCRT-III component CHMP4C has been shown to regulate the timing of exit from cytokinesis and abscission in response to activation of the Aurora B-dependent abscission checkpoint[5,13]. Its presence, however, is dispensable for this process as cells may still complete abscission after knockdown of CHMP4C (ref. 5). We hypothesized that if CHMP4C were the effector for signalling through PKCε–Aurora B–Borealin after engagement of the abscission checkpoint, loss of the protein would overcome the failure of cytokinesis we observe when signalling through this cascade is perturbed. Following knockdown of CHMP4C with each of three targeted siRNAs in the DLD1 GFP-Aurora B cell lines, we observed rescue of the binucleate cell phenotype elicited by Aurora B S227A (Fig. 4a and Supplementary Fig. 5a). Similarly, knockdown of PKCε and the associated binucleate cell phenotype could also be rescued by knockdown of CHMP4C (Fig. 4b), suggesting that exit from cytokinesis after engagement of the abscission checkpoint is being regulated by signalling through PKCε.

**Localization of CHMP4C is perturbed with PKCε inhibition.**
After abscission checkpoint activation, CHMP4C is retained on the midbody arms due to the inhibitory phosphorylation at S210 by Aurora B[5,13,15]. This prevents the translocation of CHMP4C to the midbody ring, where the final scission event is to take place. When considering the observations that S227/T232-phosphorylated Aurora B localizes to the midbody ring (Supplementary Fig. 2a) and the requirement for CHMP4C in the cytokinesis failure phenotype (Supplementary Fig. 5a) we hypothesized that the localization of CHMP4C may be influenced by signalling through PKCε.

Transiently expressed HA-CHMP4C in HeLa cells was distributed as previously described[5]. During early cytokinesis, CHMP4C is present on the midbody arms, adjacent to the midbody ring (as can be demonstrated by two peaks in the green channel on the pixel intensity profiles), whilst during late cytokinesis it is concentrated within the midbody ring (single green peak of the pixel intensity profile; Supplementary Fig. 5b). CHMP4C, which cannot be phosphorylated by Aurora B to regulate abscission timing (HA-CHMP4C S210A)[5,13], is restricted to the midbody arms, indicating that phosphorylation of this site

is required for recruitment/retention at the midbody ring. Interestingly, treatment of these cells with the PKCε inhibitor BLU577 similarly restricts the localization of CHMP4C to the midbody arms, adjacent to the midbody ring (Supplementary Fig. 5b). On transient expression of HA-CHMP4C in DLD1 (Fig. 4c) or HEK293 GFP-Aurora B-expressing cells (Supplementary Fig. 5d) we observed co-localization with Aurora B in the midbody ring. The midbody localization of CHMP4C was restricted to the midbody arms in DLD1 cells or completely lost in HEK293 cells on co-expression of Aurora B S227A, indicating that in these models localization to the midbody is dependent on the S227/T232-phosphorylated Aurora B. When HA-CHMP4C S210A was transiently expressed in these cells, it was again localized to the midbody arms, independent of PKCε and Aurora B activity or phosphorylation (Supplementary Fig. 5c,d). It has been noted elsewhere that the ESCRT machinery may be dispensable for abscission in HEK293 cells[26], however we observe clear localization of GFP-Aurora B WT and S227A to the midbody and context-dependent exogenous CHMP4C co-localization in these cells also.

Interestingly, when HA-CHMP4C (WT or mutant) was transiently expressed with GFP-Borealin in the DLD1 cell line, we again see phenocopy of the localization of CHMP4C when S165 cannot be modified (Fig. 4d). Specifically, it was observed that in GFP-Borealin (WT)-expressing cells, CHMP4C can localize to the midbody ring, while its localization is restricted to the midbody arms if GFP-Borealin (S165A) is expressed.

Capalbo et al.[13] propose that phosphorylation of CHMP4C S210 by Aurora B maintains CHMP4C in a closed configuration before abscission, therefore the localization of the S210A mutant would not be subject to regulation by the CPC, and could still localize to the midbody forming the open conformation membrane-associated polymers required. We therefore suggest that PKCε ultimately drives the localization of CHMP4C through its interaction with Aurora B and Borealin for final abscission to occur and that the absence of Aurora B S227 phosphorylation impairs exit from cytokinesis.

Together these data strongly indicate that PKCε and Aurora B control abscission timing by regulating exit from the abscission checkpoint. We propose a model in which PKCε acts directly upstream of Aurora B thus maintaining CHMP4C away from the midbody under conditions of chromatin retention in the furrow (Fig. 4e).

## Discussion

We have described an emergent role for PKCε in the final abscission checkpoint, where phosphorylation of the key mitotic regulator Aurora B is required for successful exit and completion of abscission. Aurora B and the CPC are involved in the regulation of contractile ring formation through a number of mechanisms, including modulation of RhoA activity and myosin II binding to the cytoskeleton (for review see ref. 8) but has also more recently been reported to play an active role in regulation of abscission. First, retention of active Aurora B at the midbody before abscission delays final scission. This occurs in the presence of chromatin trapped in the furrow, activating the NoCut or abscission checkpoint[1–5,15]. In human cells, this delay in abscission exit induces a stable intercellular canal at the midbody, to allow for resolution of the chromatin bridge between the two daughter cells[3]. Second, Aurora B is known to phosphorylate key components of the abscission machinery such as CHMP4C to delay abscission through preventing the relocalization of the complexes to the abscission zone[13] (Fig. 4d). Recently Petsalaki and Zachos[15] demonstrated that the

abscission checkpoint could be activated through Clk1, 2 or 4 phosphorylation of Aurora B at S331, resulting in CHMP4C S210 phosphorylation and delaying abscission until such time as the chromatin trapped in the furrow was relieved. This suggests that phosphorylation of Aurora B at S331 may provide the brake

for the abscission checkpoint, while S227 phosphorylation is the release for exit and abscission.

We show that phosphorylation of S227 of Aurora B is not required for the catalytic function of the kinase per se, rather it plays a role in its action by altering substrate selection. The

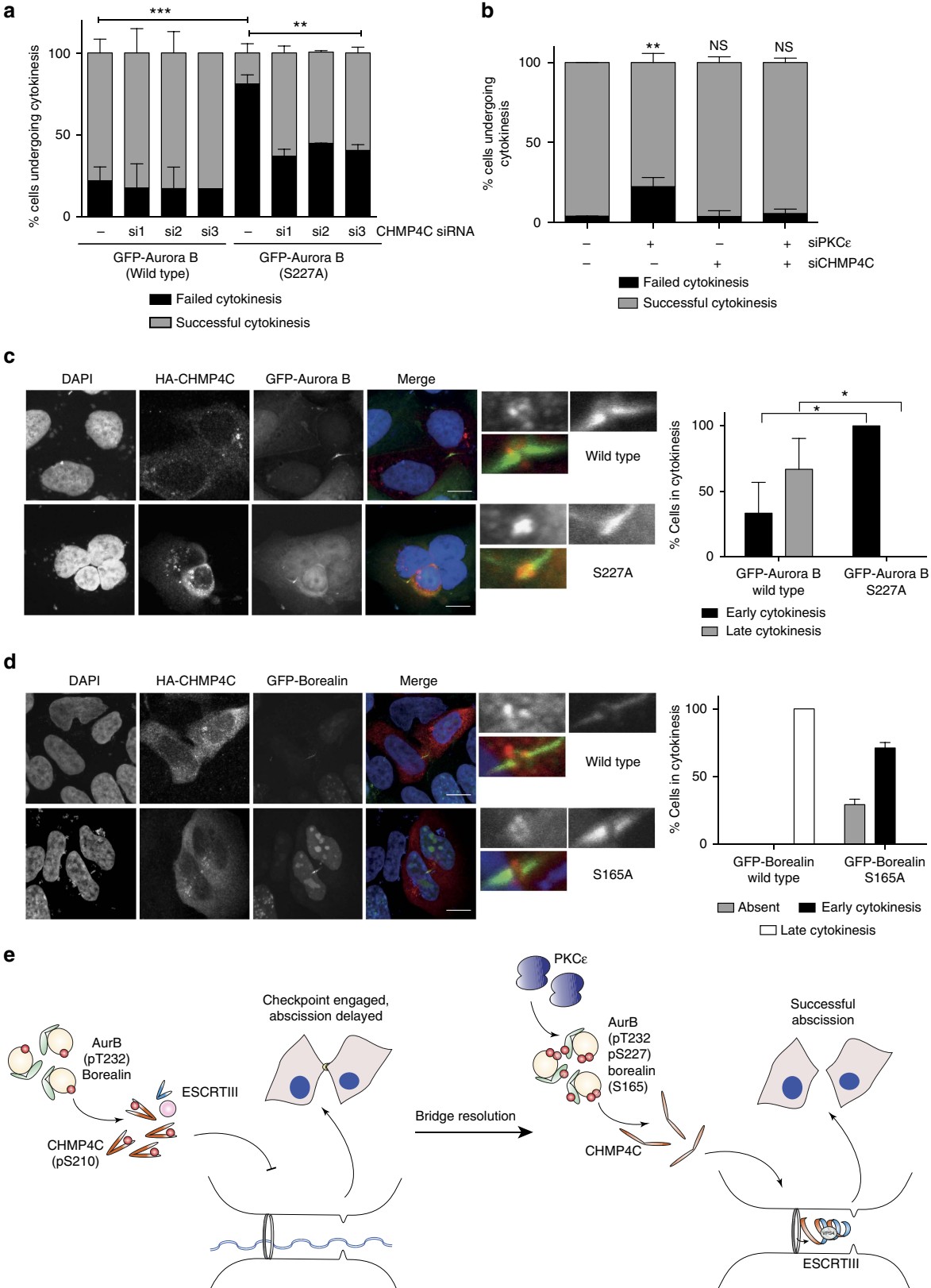

requirement for Aurora B to exist as one specific modified form to enter cytokinesis and another to exit while retaining catalytic activity is intriguing. It might be expected that the vast array of Aurora B substrates will change with the morphological and biochemical changes occurring in the cell during this time and this is reflected in the need to switch specificity. While the approach we adopted was to screen known substrates of Aurora B based on the published literature and known consensus motifs, screening different candidate sites and proteins for modification by both forms of Aurora B may prove insightful in determining substrates selected by the T232-only phosphorylated species. It has been proposed previously that Aurora B switches preference to phosphorylate serine–proline motifs on microtubule-associated substrates in cytokinesis rather than threonine–proline motifs during mitosis[11], although we have not noted any such bias from our peptide screen. In Aurora A, the corresponding serine can be phosphorylated by GSK3 to reportedly negatively regulate the kinase in *Xenopus* larvae in response to insulin and progesterone during oocyte maturation[27]; we would speculate that this may also be a change of function rather than inactivation.

The evidence indicates that when Aurora B S227 is not phosphorylated, the cell fails abscission. This may be because the cells fail to complete cytokinesis due to an inability to resolve the trapped DNA and/or failure to exit the checkpoint. We have demonstrated previously a critical role for PKCε in the response to metaphase catenation[20]. The inhibition of PKCε activity during the metaphase–anaphase transition leads to an increase in the number of anaphase ultra-fine bridges that persist through to cytokinesis, activating the abscission checkpoint. The processive changes the cell is undergoing at this time is consistent with the hypothesis that S227 phosphorylation induces a switch in substrate selection that would be required during these transitions. We and others[20,28] have demonstrated that some tumour-derived cell models have a defective G2 catenation checkpoint, meaning that they are more likely to enter mitosis with residual catenation resulting in an increase in the number of chromatin bridges as they traverse metaphase and anaphase. We see evidence of this in the Aurora B S227A mutant cells as the presence of PICH-positive structures in anaphase (Pike, unpublished observations) increasing the likelihood of residual DNA in the cytokinetic furrow, as well as the observed increase in cells attempting to undergo cytokinesis after PKCε inhibition or Aurora B S227A expression with Lap2β-positive chromatin bridges we report here. We cannot rule out an additional function for Aurora B S227 phosphorylation at the metaphase–anaphase transition in these cells, especially when considering the already reported role for PKCε and the integral role of Aurora B

during mitosis. Our results indicate however that independently of this, PKCε has a specific role in cytokinesis where it triggers the phosphorylation of Aurora B at S227.

The observed abscission failures with consequent binucleation appears to be dependent on engagement of known components of the abscission checkpoint indicating that it may be due to failure to exit from the abscission checkpoint. This suggests that this pathway is conditional on resolution of the non-disjunction errors that cause chromatin to be trapped in the furrow, phosphorylation occurring once all residual chromatin is removed from the midzone, thereby ensuring that cytokinesis is coordinated with completion of chromosome segregation. In the cell models studied, PKCε appears to be required for Aurora B phosphorylation in a high percentage of cytokinesis and failure of this results in abscission failure. It therefore appears that the abscission checkpoint is triggered in most cytokinesis. This may be due to a necessity for this process in most cell divisions in these specific cell lines reflecting the particular aneuploid state of these models.

There now appears to be a complex set of functions of PKCε in mitosis and cytokinesis. We propose a potentially pleiotropic role for PKCε at the cytokinesis furrow, controlling the previously defined RhoA–Actin signalling pathway and, as defined here, modulating Aurora B function. Evidently, PKCε at the midbody acts to control these processes in the context of the Aurora B-dependent checkpoint, engaged in sensing the conditions arising from non-disjunction and permitting completion of cell division.

## Methods

**Reagents.** All reagents were purchased from Sigma-Aldrich unless otherwise stated. BLU577 was kindly provided by Dr Jon Roffey, Cancer Research Technology, UK.

**Cell culture.** All cell lines were cultured in Dulbecco's modified Eagle's medium (Gibco) + 10% fetal calf serum and were obtained from the American Type Culture Collection unless otherwise stated. For siRNA transfections, Lullaby (OZ Biosciences) was used according to the manufacturer's recommendations; all siRNAs (Dharmacon) were used at a final concentration of 20 nM. Tetracycline-inducible DLD1 (a kind gift from Prof. Stephen Taylor) and 293 cell lines (Invitrogen) were generated using the T-Rex Flp-In system (Invitrogen) according to the manufacturer's instructions. To induce GFP-Aurora B, GFP-Borealin or GFP-PKCε expression, cells were cultured in Dulbecco's modified Eagle's medium containing 10% fetal calf serum and tetracycline (100 ng ml$^{-1}$) for 16 h before assay. Cells were treated for 30 min with inhibitors unless otherwise stated. Cell lines were routinely tested for mycoplasma.

**Kinase assay.** Assays were conducted in triplicate in 50 μl reactions containing 20 mM Tris (pH 7.5), 5 mM MgCl$_2$, 0.5 mM dithiothreitol (DTT), 0.2% Triton X-100, 100 μM ATP and 5 μCi [γ$^{32}$P]-ATP (Amersham). Reactions were incubated

**Figure 4 | PKCε and Aurora B influence CHMP4C localization during cytokinesis.** (**a**) DLD1 GFP-Aurora B cell lines were transiently transfected with three separate CHMP4C siRNA, imaged using live-cell time-lapse microscopy and scored for outcome of cytokinesis; successful cytokinesis resulting in two daughter cells or failed cytokinesis resulting in a binucleate cell. Graph represents the mean ( ± s.e.m.) of three independent experiments; a minimum of 50 cells were counted per experiment. Two-way analysis of variance (ANOVA), ** $= P \leq 0.01$, *** $= P \leq 0.001$. (**b**) HeLa cells were transfected with PKCε siRNA, a pool of three CHMP4C siRNA or PKCε siRNA and the pool of CHMP4C siRNA, imaged using live-cell time-lapse micrsocopy and scored for outcome of cytokinesis as per the criteria above. Graph represents the mean ( ± s.e.m.) of three independent experiments; a minimum of 50 cells were counted per experiment. Two way ANOVA, not significant (NS) $= P > 0.05$, ** $= P \leq 0.01$. (**c**) DLD1 GFP-Aurora B WT and S227A cell lines (green) were transiently transfected with HA-CHMP4C (red) to look for midbody localization during cytokinesis. Cells were scored for the presence or absence of HA-CHMP4C at the midbody. A minimum of 12 high-resolution, single-cell images per condition from four experiments were acquired; a representative image is shown here. Scale bar, 10 μm. Student's *t*-test, * $= P \leq 0.05$. (**d**) DLD1 GFP-Borealin WT and S165A cell lines (green) were transiently transfected with HA-CHMP4C (red) to look for midbody localization during cytokinesis. Cells were scored for the presence or absence of HA-CHMP4C at the midbody. A minimum of five high-resolution, single-cell images per condition from two experiments were acquired; a representative image is shown here. Scale bar, 10 μm. (**e**) Working model: chromatin trapped in the cytokinesis furrow engages the Aurora B-dependent abscission checkpoint. We propose that the association through T232 only phosphorylated Aurora B and Borealin, CHMP4C is maintained in a S210 phosphorylated, closed, inactive conformer distal to the midbody. On bridge resolution, PKCε phosphorylates Aurora B on S227, which in turn phosphorylates Borealin S165, allowing for CHMP4C to assume an open, active conformation to polymerize with other ESCRT-III components and facilitate successful abscission.

for 10 min at 30 °C using 100 ng of each kinase per reaction. Reactions were terminated by spotting onto P81 cellulose paper or addition of 4 × lysis buffer (Invitrogen).

Cold kinase assays were conducted in 50 µl reactions containing 20 mM Tris (pH 7.5), 5 mM MgCl₂, 0.5 mM DTT, 0.2% Triton X-100 and 100 µM ATP. Reactions were incubated for 30 min at 30 °C using 1 µg of each recombinant protein per reaction. Recombinant Borealin protein was purchased from Abcam (ab107144). Reactions were terminated by addition of NuPAGE 4 × LDS sample buffer (Thermo Fisher) before western blotting with the appropriate phospho-antibodies to detect kinase activity against specific phospho-sites.

**Thermal shift assay.** Thermal shift assays typically followed the protocol as described in ref. 29. Briefly, 100 µl of protein ($\sim 5$ µg) in 50 mM HEPES (pH7.6), 300 mM NaCl, 5% glycerol and 1 mM DTT was incubated with 20 mM MgCl₂ and 0–10 µM ATP for 30 min at 4 °C in the presence of 5 × Sypro Orange dye (Sigma-Aldrich). From the 100 µl reaction mixture per condition, 4 × 20 µl was loaded onto RT–PCR plates in technical quadruplicates. Melting curves were then assessed in an Applied Biosystems 7500 Fast RT–PCR system (Life Technologies). Temperature was cycled up from 25 to 95 °C in 1 °C min⁻¹ increments, with measurements taken every 0.5 °C.

Curves were trimmed manually and a Boltzmann Sigmoidal curve was fitted to the data in GraphPad Prism. The inflection point of the curve, $T_m$, was taken from all conditions. The average $T_m$ value of the untreated control replicates was subtracted from the values of the treated wells to obtain the difference in $T_m$ caused by ATP binding, termed the $\Delta T_m$ value.

**Peptide array.** Peptides predicted to be PKCε or Aurora B substrates were identified and 15mers arrayed on nitrocellulose by the Peptide Synthesis laboratory of the Francis Crick Institute. Membranes were blocked in 0.2 mg ml⁻¹ BSA, 20 mM Tris (pH7.5) and 0.02% Tween-20 overnight. Membranes were subsequently incubated with the appropriate recombinant protein (PKCε kinase domain 5 µg, Aurora B WT 100 µg and Aurora B S227A 100 µg) and 10 mM MgCl₂ and 100 µM [γ³²P]-ATP (5 µCi ml⁻¹) for 10 min followed by extensive washing in H₂O and acetic acid. Membranes were then exposed to film before spot intensity analysis using the ImageQuant TL7 (GE Lifesciences).

**Proximity ligation assay.** Cells were grown on eight-well-chambered slides (Falcon) and fixed as for immunofluorescent imaging. Proximity ligation assay was conducted using a kit (Sigma) as per the manufacturer's instruction using the antibody pairs anti-INCENP (rabbit) (Abcam ab12183), anti-Aurora B (mouse; BD AIM1 no. 611082) and anti-Aurora B, anti-PKCε (rabbit; Abcam clone EPR1482 ab124806). Nonspecific IgG (rabbit sc-2027 and mouse sc-2025; Santa Cruz) were used in conjunction with anti-Aurora B and anti-PKCε (respectively) as negative controls to demonstrate the specificity of the proximity ligation assay reaction.

**Constructs.** Aurora B cDNA was a kind gift from Dr Mark Petronczski and was cloned into pcDNA5/FRT/TO (Invitrogen) engineered to express an N-terminal GFP using the InFusion cloning kit (BD) according to the manufacturer's instructions. For recombinant protein production, pET-Duet-1 Aurora B:INCENP was a gift from Dr Jon Elkins, University of Oxford. RFP-Lap2B was purchased from Addgene. Site-directed mutagenesis was completed using QuickChange mutagenesis kit (Agilent) according to the manufacturer's instructions. All clones were sequence verified.

**Microscopy.** For live-cell time-lapse microscopy, cells were cultured on LabTek chambered coverglass slides (Nunc) in Leibovitz CO₂-independent media (Gibco). A low light level inverted microscope (Nikon TE2000) imaging system equipped with a laminar flow heater to maintain a constant temperature of 37 ± 0.01 °C, a PlanFluor × 40 differential interference contrast (DIC) lens and a Xenon lamp for fluorescent excitation. Images were taken using a high-quantum efficiency charge-coupled device camera (Andor Ixon) every 5 min. Still images were taken using an inverted laser scanning confocal microscope (Carl Zeiss LSM 780) equipped with a × 63 Plan-APOCHROMAT differential interference contrast oil-immersion objective.

**Immunofluorescence and immunoblotting.** For immunofluorescence experiments, cells were grown on 13 mm glass coverslips and were simultaneously fixed and permeabilized in PTEMF buffer (4% paraformaldeyhide, 0.2% Triton X100, 20 mM PIPES (pH 6.8), 10 mM EGTA and 1 mM MgCl₂) for 30 min. The following primary antibodies were used in these assays are as follows: mouse anti-Aurora B (AIM-1,611082, BD, 1:500); rabbit anti-Aurora B phosphoThr232 (TA325250, Origene, 1:300); rabbit anti-Aurora B phosphoSer227 (made in-house, 1:100); mouse anti-alpha tubulin (clone DM1A, T9026, Sigma, 1:1,000); rabbit anti-INCENP (ab12183, Abcam, 1:300); mouse anti-Borealin (ab67126, Abcam, 1:300); mouse anti-Lap2b (611000, BD, 1:300); and mouse anti-HA.11 (clone 16B12, MMS-101P, Covance, 1:300). Primary antibodies were detected with Alexa

Fluor-conjugated secondary antibodies (Life Technologies). All coverslips were mounted using ProLong Diamond with DAPI (Invitrogen).

For immunoblotting, cells were lysed in 1 × NuPAGE LDS sample buffer (Thermo Fisher Scientific) and sonicated for 3 × 10 s on ice. Proteins were separated by SDS–PAGE and transferred to polyvinylidene difluoride membranes (Millipore). The following primary antibodies were used: mouse anti-Aurora B (AIM-1,611082, BD, 1:2,000); rabbit anti-Aurora B phosphoThr232 (TA319253, Origene, 1:1,000); rabbit anti-Aurora B phosphoSer227 (made in-house, 1:1,000); mouse anti-alpha tubulin (clone DM1A, T9026, Sigma, 1:10,000); rabbit anti-Borealin phosphoSer165 (made in-house, 1:500); mouse anti-Borealin (ab67126, Abcam, 1:2,000); rabbit anti-Histone H3 (9715, Cell Signaling, 1:5,000); rabbit anti-Histone H3 phosphoSer10 (9,701, Cell Signaling, 1:2,000); rabbit anti-CHMP4C (GTX122876, Genetex, 1:1,000); and mouse anti-Histone H2AX phosphoSer139 (clone JBW301 no. 05-636, Millipore, 1:5,000). Mouse and rabbit horseradish peroxidase-conjugated secondary antibodies (NA931V and NA93V, GE Lifesciences) were used at appropriate dilutions. Chemiluminescence was detected using Luminata Classico western horseradish peroxidase substrate (Millipore) and imaged using the ImageQuant 4000 mini (GE Lifesciences). Band densitometry was carried out using Image J software and normalized to a loading control. Whole membranes are presented in Supplementary Fig. 6.

**Expression and purification of Aurora B/INCENP.** A pET-Duet vector encoding human Aurora B (WT and S227A) and human INCENP WT (residues 837–918) was transformed into electrocompetent Rosetta pLysS *Escherichia coli* cells. A 10 ml starter culture was used to inoculate 500 ml of LB medium containing 100 µg ml⁻¹ ampicillin. The culture was grown at 37 °C to an OD₆₀₀ nm = 0.7. The cultures were removed from the incubator and allowed to cool to room temperature. Subsequently, isopropyl-β-D-thiogalactoside was added to a final concentration of 1 mM and the cultures grown overnight at 20 °C. The cells were pelleted by centrifugation (2,000g for 10 min) and re-suspended in 25 ml of lysis buffer (25 mM Tris (pH 8.0), 1 M NaCl, 1 mM EDTA and 1 mM DTT) with the addition of a Roche Protease Inhibitor Cocktail Tablet. The cells were lysed by a combination of lysozyme addition and 3 × 10″ sonication bursts. Subsequently, polyethyleneimine (25 kDa, linear) was added to a final concentration of 0.15% (w/v) and incubated on ice for 30 min. Finally, the insoluble debris (and polyethyleneimine–DNA complexes) was precipitated by centrifugation at 20 K for 20 min in an ultracentrifuge and filtrated before purification using an AKTA Pure System (GE Healthcare) with a 1 ml HisTrap column attached. The running buffer was 25 mM Tris (pH 8.0), 200 mM NaCl, 1 mM EDTA and 1 mM DTT. The same buffer supplemented with 500 mM imidazole was used for elution. The proteins in the peak were pooled and subjected to size exclusion using a Superdex200 10/300 column. Peaks containing both the Aurora B as well as the INCENP subunit were pooled, concentrated and flash-frozen in liquid N₂ until use. Recombinant Aurora B WT and S227A proteins were assessed for their phosphorylation state (pS227 and pT232) and kinase activity against the substrate Histone H3 S10 before use in all assays. At low efficiency of production of Aurora B, T232 but not S227 is phosphorylated. However, on highly efficient production (that is, high concentration) as determined using the above protocol, both sites become phosphorylated.

**Expression and purification of PKCε kinase domain.** The cDNAs encoding human PDK1 and a fusion protein consisting of glutathione S-transferase (GST)-3C-PKCε (human, kinase domain) were inserted into the two multiple cloning sites of the pFL vector (kind gift from Prof. Imre Berger, University of Bristol). High-titre baculovirus stocks were generated using standard Bac-to-bac (Invitrogen) protocols using Sf21 cells. For protein expression, 500 ml of Sf21 cells at 1 × 10⁶ cells per ml grown in SF900-III medium (Life Technologies) were infected with a multiplicity of infection of 1 for 72 h. The infection process was monitored by cell counts and measurements of the insect cell diameter. The cells were collected by centrifugation (2,000g for 10 min) and re-suspended in 25 ml of GST-binding buffer (25 mM HEPES, (pH 7.5), 125 mM NaCl, 1 mM DTT and 1 mM EDTA) supplemented with a protease inhibitor tablet (Roche). The lysate was loaded onto a 5 ml GSTrap column (GE Healthcare) at 1 ml min⁻¹, and GST-3C-PKCε was eluted in binding buffer with 30 mM reduced glutathione. The fusion protein was cleaved overnight at 4 °C with prescission protease. The cleavage mixture was loaded onto a Resource15Q column at 1 ml min⁻¹ and a linear NaCl gradient (50–500 mM) was applied. The PKCε kinase domain eluted as a single peak around 300 mM. The kinase-containing fractions were pooled and subjected to a final size-exclusion step using a Superdex 75 in a buffer containing 25 mM HEPES (pH 7.5), 125 mM NaCl and 1 mM TCEP.

**Statistical analysis.** For experiments where the data include more than two conditions, a two-way analysis of variance using multiple comparisons was used, in all other cases an unpaired *t*-test was used for analysis. Prism software (Graphpad) was used for all calculations. The level of statistical significance is represented as follows: not significant = $P > 0.05$, * = $P \le 0.05$, ** = $P \le 0.01$, *** = $P \le 0.001$ and **** = $P \le 0.0001$.

**Data availability.** The authors declare that the data supporting the findings of this study are available within the article and its Supplementary Information files, or are available from the authors on request.

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

## Acknowledgements

This work was supported by the Francis Crick Institute, which receives its core funding from Cancer Research UK (FC001130), the UK Medical Research Council (FC001130) and the Wellcome Trust (FC001130). We thank all of the Francis Crick Institute core facilities for valuable support throughout this project, in particular the light microscopy, peptide synthesis and protein production facility. We also thank Stephen Taylor for kindly supplying the tetracycline inducible DLD1 cell line. We acknowledge Dr Mark Petronczski for providing the Aurora B construct, Dr Jon Elkins for the Aurora B construct to produce the recombinant protein and we thank both for their helpful advice and discussions. Thanks also to Drs Katharina Deiss and Philippe Riou for critical reading of the manuscript.

## Author contributions

T.P., N.B., J.C. and P.J.P. devised experiments; T.P. carried out experiments; S.K. prepared and purified recombinant proteins. All authors contributed to writing the manuscript.

## Additional information

**Competing financial interests:** The authors declare no competing financial interests.

**Publisher's note**: 

