## [Peer Review File · Nature Communications]

Reviewers' Comments:

Reviewer #1 (Remarks to the Author)

The paper by Pike et al explores the regulation of Aurora B kinase by PKC ϵ in cytokinesis. PKC ϵ was previously thought to have a role in this process only in a subset of transformed cells. The authors claim that the abscission checkpoint is directly modulated by phosphorylation of Aurora B by PKC ϵ . Phosphorylation at Ser227 would change the specificity of Aurora B for a number of substrates including Borealin, a subunit of the CPC that plays a crucial role in abscission. In addition, the localization of the ESCRTIII component CHMP4C is described to be lost upon inhibition of PKC ϵ or co-expression of the mutant Aurora B S227A.

The results described in this manuscript are potentially quite interesting and the authors' conclusions could contribute significantly to the understanding of the abscission checkpoint. I have no doubt that the study would be of interest to Nature Communications readership. However the main conclusions are not fully supported at present by the evidence shown here. In particular, the quality and resolution of the images shown throughout is not enough to support the claims made. Due to the nature of the study, it is essential that this issue is addressed before publication. Microscopy images should show individual cells; different channels need to be shown separately in black and white; high magnification insets of the midbodies or other subcellular structures need to be shown, and specific structures labeled properly. In addition, more attention should be paid to accurate description of the statistical analysis, including tests applied, etc.

Specific points:

- Aurora B Ser 227 residue is identified (among others) as a PKC ϵ substrate, and shown to be such in vitro. To explore the possible roles of this phosphorylation in vivo, the authors start by trying to show that the proteins colocalize in cells. Figure Suppl 1C only shows localization of exogenous tagged protein, very low resolution and very dispersed. The cells are clumped and it is difficult to assess the extent of colocalization. The authors should use antibodies against the endogenous proteins when possible. In addition, the figure legend is incomplete (no mention to experiment in lower panel, is this an analog sensitive kinase?). Figure 1E shows similar problems: in addition, the anti phosphoepitope specific antibody (anti-Aurora pSer227) shows a dispersed nuclear staining that the authors claim is non-specific. This claim would require further validation (using RNAi for instance). Once the antibody has been fully validated, it should be used for a detailed analysis of the time when this phosphorylation is present at the different stages of late

mitosis/cytokinesis.

- The PLA experiments are not described in full detail. Please specify the pairs of primary antibodies used in each experiment (explaining clearly if antibodies against the tags rather than against the endogenous proteins are used). The authors need to show the localization of the endogenous proteins (in supplemental materials). In figure S1D GFP-PKC ϵ is not present at the midbody in the lower panel. In general, the PLA experiments are lacking negative controls and the signals need to be quantified.
- In Supplemental Figure 2 the panels not always correspond to the legend, it is incredibly confusing. In panel 2A: GFP constructs do not seem to be expressed at "near endogenous levels" as stated in figure legend. Show quantification of endogenous/exogenous pools of protein.
- Figure Suppl2C legend (B?): please define the criteria used to calculate these times, they seem unusually long.
- Figure 2 LAP2beta is typically used to make it easier to detect chromatin in the furrow during live imaging. The LAP2 beta staining (fixed cells) shown in figure 2C is almost impossible to visualize. Similar comments/advice as in previous experiments with regards to this figure. Quantification of the signal in live cells would not have been essential in this case (it could have been done more efficiently in fixed cells).
- The panels in Supplemental figure 2F are very low quality and uninformative. The localization of GFP-Aurora B in early mitosis does not look normal. Again, the authors need to show higher magnification insets, black and white panels, etc. It would be nice to show colocalization with other CPC components in all stages of mitosis. It is important to do a similar analysis with Aurora B S227A and describe the differences in detail. In Supplemental figure 2G, INCENP does not show correct localization. Please show colocalization with CPC components (including Survivin, commercial antibodies are available for this kind of analysis) in different stages of mitosis and cytokinesis.
- Supplemental Figure 3B is lacking specific details of statistical analysis, test used, p values, etc.
- In figure 3, the analysis of Borealin mutant is done in cells transiently transfected. This introduces variability -from different levels of expression. The images do not have very good resolution, it is difficult to visualise Borealin -some frames are out of focus, the cells vacuolised. The localization of the mutant needs to be described more accurately. In panel C the statistical analysis is missing.
- The localization of Aurora B pSer227 specifically in the midbody ring would be potentially of great interest, but as I mentioned before the antibody needs to be better validated before reaching any conclusion.
- The different localizations of HA-CHMP4C look all the same at the level of resolution shown in S4C (despite the insets). The authors need to show the endogenous protein when possible for reference.

Reviewer #2 (Remarks to the Author)

The manuscript "PKC ϵ switches Aurora B specificity to exit the abscission checkpoint" by Pike and colleagues shows a novel pathway that regulates abscission checkpoint exit. As the authors claimed PKC ϵ phosphorylates Aurora B at S227, which switches Aurora B's kinase specificity against several substrates including Borealin and ESCRT-III component CHMP4C, then facilitates abscission exit. Generally, this manuscript is of interest in the cell cycle field and is helpful for people to understand the cell division more precisely. However, some evidences the authors presented are not convincing. These problems summarized below should be clearly solved before consideration for acceptance of this manuscript.

Major problems:

1. No rescue experiments. Since the authors have claimed that in yeast glutamic acid could potentially phospho-mimetic Aurora B phosphorylation at S227 in eukaryotes, they should carry this rescue experiment in DLD1 or HeLa cells to see whether the cells expressing Aurora B S227E (or S227D) mutant proteins separate normally during cytokinesis. They also need to do the rescue experiments with Borealin and CHMP4C.
2. The authors use Proximity Ligation Assay to prove that PKC ϵ and Aurora B interacted in midbody(Fig1D), however, considering the limited space of midbody and normally very few PKC ϵ localizes in midbody(Figure S1A,S1B), this experiment is not very convincing without negative controls. It might be better for the authors to find out the interaction domain of PKC ϵ with Aurora B, and use the domain-depletion construct to perform this experiment.
3. Similar with the question above, the timing of Aurora B S227 phosphorylation by PKC ϵ should be carefully checked in the whole cell cycle. The S227 fluorescence staining and WB analysis with cells in different cell cycle stages are necessary. Moreover, the model (Fig. 4D) proposes that upon bridge resolution, PKC ϵ phosphorylates Aurora B, the authors should give more direct evidence to prove this conclusion.
4. The authors claim that double phosphorylated Aurora B has greater kinase activity to Borealin. However, this experiment procedure description is too simple, making the result of the peptide array to be difficult to be understood (Fig.3A). The authors should make it clear that how double-phosphorylated Aurora B WT and single-phosphorylated S227A recombinant proteins were prepared and whether they were active.
5. The logical relationship between Fig.3 and Fig.4 is not strong. The functions of Borealin S165 phosphorylation and Aurora B S227 phosphorylation toward CHMP4C needs more direct evidence.

Minor problems:

1. Fig.1C: no loading control of PKC ϵ . ATP should be labeled as ^{32}P -ATP.
2. Fig.S2A: no loading control of each sample and it's hard to understand why endogenous Aurora B has two lanes. Furthermore, GFP-Aurora B

seems at least 5 times more than endogenous level, not just near the level as authors claimed.

3. FigS2B, Fig.4B and 4C: the cells counted in each experiment should be shown.

4. FigS2E: the time of each still image should be shown.

5. Fig.2D and Fig.3C: Only two independent experiments were carried out, how could the authors get the significant difference?

6. Fig. S3A and S3B: The total loading amount of Aurora B is inconsistent. It seems S277A is much more than WT, thus the conclusion that the two proteins possess equal activities is not convincing.

Reviewer #3 (Remarks to the Author)

A. Summary of the key results

PKC ϵ has previously been found to have roles important to resolution of concatenated chromosomes as cells exit mitosis and to the function of ZO-1 and RhoA during cytokinesis. Here, further insight into PKC ϵ function is obtained through investigation of its regulatory interface with the Aurora B kinase.

To find substrates of PKC ϵ when it localizes to the intercellular bridge, a peptide array biased toward midbody proteins with potential recognition sites for phosphorylation was employed. One hit from this screen was Aurora B S227, a conserved residue near the auto-phosphorylation site T232. An S227 phospho-specific antibody was raised and used to confirm that PKC ϵ phosphorylates Aurora B. Results suggest that phosphorylation of S227 does not affect the phospho-status of T232. Despite this, expression of Aurora B S227A resulted in an increase in binucleated cells. This was confirmed by live-imaging experiments showing that expression of Aurora B S227A resulted largely in failed cytokinesis, similar to treatment with BLU577, a PKC ϵ inhibitor. In many cases, cells expressing Aurora B S227A had visible chromatin bridges. Yet, the timing of the midbody-stage did not seem to change under these conditions (authors should clarify, this was measured in the minority of cases where midbodies were resolved -or does it include the timing of midbody regression).

Finding that intracellular targeting, catalytic activity, and ATP binding of Aurora B are not altered by phosphorylation of S227, the authors pursue the hypothesis that substrate specificity is modulated by this phospho-modification. To test this, a second peptide array is used (biased toward Aurora B substrate sites and comparing S227A to Aurora B capable of double phosphorylation). One of several positive (differential) hits in the above screen is tested functionally: site S165 in the CPC subunit Borealin. Expression of this mutant phenocopies the high rate of abscission failure observed when Aurora B S227A is expressed or when PKC ϵ is inhibited. The authors conclude that PKC ϵ is required to phosphorylate Aurora B

when there is DNA retained in the cleavage furrow, which in turn must phosphorylate Borealin at 165 in order to trigger completion of abscission -or to trigger an abscission checkpoint, allowing time for resolution of lagging DNA (but see critique below).

Closer inspection of phosphorylated Aurora B led to the appreciation that when phosphorylated at S227, Aurora B is exclusively at the central midbody ring. This localization pattern suggested a role in modulation of the abscission machinery, which is thought to be regulated at this site. Looking specifically at CHMP4C, the authors find that treatment with BLU577 or expression of Aurora B S227A alters CHMP4C targeting at the midbody (this is clear in the examples in Fig 4A, but not as easily seen in supplemental Fig 4).

Although the logic does not seem entirely clear-cut, the authors next test whether knockdown of CHMP4C would rescue abscission defects seen when Aurora B S227A is expressed. When the ability to execute an abscission checkpoint is abrogated by depletion of CHMP4C, cells no longer fail as frequently in abscission following Aurora B S227A expression. At face value, this means that abscission failure that results from lack of PKC ϵ signaling via Aurora B modification is due to stimulation of an abscission checkpoint coupled with an inability to progress forward from there or to sustain the checkpoint-arrested state.

B. Originality and interest:

The results presented are original and will be of high interest to the field.

C. Data & methodology:

One main question is about the proteins used to compare Aurora B to its S227A counterpart. While the relevant consideration is the effect of phosphorylation at 227 on binding and activity, the Methods describes the use of recombinant protein without clarifying how S227 is phosphorylated. i.e., is there a pre-incubation with PKC ϵ ? If so, how quantitative is this phosphorylation? (if only a minor proportion is modified, this too makes the comparison difficult) How is PKC ϵ removed?

One important conclusion is the change in substrate specificity of Aurora B when phosphorylated at S227. Specific aspects of this conclusion need to be more rigorously addressed. First, clarify how many times the peptide array was performed. Second, test changes in activity towards substrates using in vitro kinase assays. Additional experiments are required to convincingly demonstrate a switch in specificity for Aurora B and its dependence on PKC ϵ . The data show that phosphorylation of S165 in Borealin is important, but whether this strictly depends on PKC ϵ is not conclusive.

Other points:

Figure S1: indicate which is Aurora B on the heat-map; what does the arrowhead indicate?

-Include more controls for the PLA (i.e., a negative control for background signal)

-Explain the M486A mutation used

-Is the BIM inhibitor the same as BLU577/compound 18? If not, why are different inhibitors used in vitro and in vivo?

-Part E, This signal is not very compelling and is based on single images (presumably the best). Is there a way to quantify?

Referring to the site "being occupied" is confusing (top of third page).

Figure S4C -the classification of early and late cytokinesis is not clear, nor is the "arm" vs. midbody ring in the images. This needs to be presented more clearly (use arrows) and in a more quantitative manner.

D. Appropriate use of statistics and treatment of uncertainties

See above for comments that have to do with quantification and reproducibility.

E. Conclusions: robustness, validity, reliability

Several novel findings are presented in this manuscript, but there are certain results that need to be more rigorously established.

F. Suggested improvements: experiments, data for possible revision

In addition to addressing the points brought up above (such as quantitative assessment of change in phosphorylation specificity for Aurora B), one aspect of this manuscript that was challenging was the lack of integration with previous findings on the role of PKC ϵ . For instance, the high rate of Lap2 β positive bridges when Aurora B S227A is expressed seems likely to reflect a role of this signaling pathway in preventing concatenation, consistent with previous reports for PKC ϵ function. This same pathway appears to be important to resolving the bridge in coordination with an abscission checkpoint.

When explaining Borealin as a downstream target of PKC ϵ -Aurora B signaling, it was suggested that cells could not respond to DNA in the cleavage furrow when the phospho-site in Borealin was mutated --but, it was unclear why so many cells would have DNA in the cleavage furrow to begin with (87% fail in cytokinesis).

Finally, to test the working model that CHMP4C depletion allows cells to progress forward despite chromatin bridges (which, again, are elevated when PKC ϵ is inhibited), downstream events such as DNA damage and chromosomal instability would be predicted to be prevalent and the manuscript would be strengthened by their assessment.

G. References: appropriate credit to previous work?

Seemed appropriate

H. Clarity and context

See problems discussed with integrating previous results and making the logic of the CHMP4C experiment more clear. A different discussion point that should be mentioned is the limitation of using known phospho-sites to screen for altered specificity of Aurora B.

Reviewer #1 (Remarks to the Author):

The paper by Pike et al explores the regulation of Aurora B kinase by PKC ϵ in cytokinesis. PKC ϵ was previously thought to have a role in this process only in a subset of transformed cells. The authors claim that the abscission checkpoint is directly modulated by phosphorylation of Aurora B by PKC ϵ . Phosphorylation at Ser227 would change the specificity of Aurora B for a number of substrates including Borealin, a subunit of the CPC that plays a crucial role in abscission. In addition, the localization of the ESCRTIII component CHMP4C is described to be lost upon inhibition of PKC ϵ or co-expression of the mutant Aurora B S227A.

The results described in this manuscript are potentially quite interesting and the authors' conclusions could contribute significantly to the understanding of the abscission checkpoint. I have no doubt that the study would be of interest to Nature Communications readership. However the main conclusions are not fully supported at present by the evidence shown here. In particular, the quality and resolution of the images shown throughout is not enough to support the claims made. Due to the nature of the study it is essential that this issue is addressed before publication. Microscopy images should show individual cells; different channels need to be shown separately in black and white; high magnification insets of the midbodies or other subcellular structures need to be shown, and specific structures labeled properly. In addition, more detail should be paid to accurate description of the statistical analysis, including tests applied, etc.

Specific points:

Aurora B Ser 227 residue is identified (among others) as a PKC ϵ substrate, and shown to be such in vitro. To explore the possible roles of this phosphorylation in vivo, the authors start by trying to show that the proteins colocalize in cells. Figure Suppl 1C only shows localization of exogenous tagged protein, very low resolution and very dispersed. The cells are clumped and it is difficult to assess the extent of colocalization. The authors should use antibodies against the endogenous proteins when possible.

- Figure S1C has been revised and described in text. DLD1 cells were stained for endogenous PKC ϵ and Aurora B in the presence or absence of BLU577.

In addition, the figure legend is incomplete (no mention to experiment in lower panel, is this an analog sensitive kinase?).

- Figure legend has now been revised and experiment with PKC ϵ M486A mutant has been removed from the manuscript for clarity

Figure 1E shows similar problems: in addition, the anti phosphoepitope specific antibody (anti-Aurora pSer227) shows a dispersed nuclear staining that the authors claim is non-specific. This claim would require further validation (using RNAi for instance).

- Figure S1E has been included to demonstrate the staining of Aurora B pS227 Aurora B siRNA treated DLD1 cells. Staining of pS227 remains nuclear however is seen to be lost from the midbody with siRNA knockdown. Western blot analysis confirms loss of Aurora B by siRNA. It should also be

noted that the pS227 antibody does not detect Aurora B S227A as we demonstrate loss of signal at the midbody in cells expressing this mutant, further validating the specificity of this antibody.

Once the antibody has been fully validated, it should be used for a detailed analysis of the time when this phosphorylation is present at the different stages of late mitosis/cytokinesis.

- Figure S1F demonstrates localization of pS227 species of Aurora B through the preceding stages of mitosis – metaphase, anaphase and telophase

- The PLA experiments are not described in full detail. Please specify the pairs of primary antibodies used in each experiment (explaining clearly if antibodies against the tags rather than against the endogenous proteins are used).

- Experimental details have been clarified in methods and figure legends.

The authors need to show the localization of the endogenous proteins (in supplemental materials). In figure S1D GFP-PKC ϵ is not present at the midbody in the lower panel. In general, the PLA experiments are lacking negative controls and the signals need to be quantified.

- Experiments have been repeated using antibodies against the endogenous proteins (Figure 1D) and the negative controls are now included in the supplementary materials (Figure S1D). Quantification of the signal at midbody is confounded by the size of the structure with the size or number of dots not necessarily a reflection of the number of molecules interacting with one another. There was clear localization to the midbody of the PLA signals with either the INCENP and Aurora B or PKC ϵ and Aurora B combination of antibodies which was absent in the negative control.

- In Supplemental Figure 2 the panels not always correspond to the legend, it is incredibly confusing.

- This has been amended and the figure panels and figure legends are now matched.

In panel 2A: GFP constructs do not seem to be expressed at "near endogenous levels" as stated in figure legend. Show quantification of endogenous/exogenous pools of protein.

- Western blots are now more accurately described in the text.

- Figure Suppl2C legend (B?): please define the criteria used to calculate these times, they seem unusually long.

- The time taken to complete cell division was defined as the time at telophase onset (bundling of microtubules at the midbody, 'dumbbell' morphology) until complete breakdown of the Aurora B/tubulin positive midbody. The timing of cytokinesis/failure of cytokinesis in this assay is comparable to what Carlton *et al.* (2012) report: abscission times of 116 \pm 45 and 137 \pm 61 minutes in control HeLa cells expressing GFP. We report here (in the GFP-Aurora B/mCherry-tubulin DLD1 cells) abscission times of 155.2 \pm 53.57 minutes for WT expressing cells and 173.3 \pm 54.24 minutes for S227A expressing cells.

- Figure 2 LAP2beta is typically used to make it easier to detect chromatin in the

furrow during live imaging. The LAP2 beta staining (fixed cells) shown in figure 2C is almost impossible to visualize. Similar comments/advice as in previous experiments with regards to this figure. Quantification of the signal in live cells would not have been essential in this case (it could have been done more efficiently in fixed cells).

- Immunofluorescent images depicting the Lap2 β positive chromatin bridges have been replaced with clearer black and white images to demonstrate the presence of these bridges in the absence of S227 phosphorylation. The quantification of Lap2 β positive bridges in live cells undergoing cytokinesis is a more accurate reflection of how these cells undergo mitosis and cytokinesis in real time after induction of each exogenously expressed protein and/or inhibitor treatment when compared to scoring cells after a defined period of time in a fixed sample.

- The panels in Supplemental figure 2F are very low quality and uninformative. The localization of GFP-Aurora B in early mitosis does not look normal. Again, the authors need to show higher magnification insets, black and white panels, etc.

- New representative images demonstrating the subcellular localization of the GFP-Aurora B constructs have been included.

- It would be nice to show colocalization with other CPC components in all stages of mitosis. It is important to do a similar analysis with Aurora B S227A and describe the differences in detail. In Supplemental figure 2G, INCENP does not show correct localization. Please show colocalization with CPC components (including Survivin, commercial antibodies are available for this kind of analysis) in different stages of mitosis and cytokinesis.

- Images demonstrating the localization of all components of the CPC with expression of the GFP-Aurora B WT or S227A mutant or inhibition of PKC ϵ are shown in Supplementary Figure 3.

Supplemental Figure 3B is lacking specific details of statistical analysis, test used, p values, etc.

- The details of the statistical analysis has now been included for this figure

- In figure 3, the analysis of Borealin mutant is done in cells transiently transfected. This introduces variability -from different levels of expression. The images do not have very good resolution, it is difficult to visualise Borealin -some frames are out of focus, the cells vacuolised. The localization of the mutant needs to be described more accurately. In panel C the statistical analysis is missing.

- Stable, inducible cell lines (DLD1 FRT-Trex cells) were created and the analysis herein included in Figure 3C
- Confocal images representing the localization of GFP-Borealin WT or S165A mutant have been included in Figure 3B
- Analysis of panel C has been included.

-The localization of Aurora B pSer227 specifically in the midbody ring would be potentially of great interest, but as I mentioned before the antibody needs to be better validated before reaching any conclusion.

- As described above, the pSer227 antibody has been further validated
- Figure S1E has been included to demonstrate the staining of Aurora B pS227 Aurora B siRNA treated DLD1 cells. Staining of pS227 remains nuclear as demonstrated by the wide view (upper panels) of cells, however is seen to be lost from the midbody with siRNA knockdown (lower panel). Western blot analysis confirms loss of Aurora B by siRNA. It should also be noted that the pS227 antibody does not detect Aurora B S227A as we demonstrate loss of signal at the midbody in cells expressing this mutant, further validating the specificity of this antibody.

- The different localizations of HA-CHMP4C look all the same at the level of resolution shown in S4C (despite the insets). The authors need to show the endogenous protein when possible for reference.

- We have improved the quality of the insets, however have been unable to detect the endogenous protein with the antibodies available.

Reviewer #2 (Remarks to the Author):

The manuscript "PKC ϵ switches Aurora B specificity to exit the abscission checkpoint" by Pike and colleagues shows a novel pathway that regulates abscission checkpoint exit. As the authors claimed PKC ϵ phosphorylates Aurora B at S227, which switches Aurora B's kinase specificity against several substrates including Borealin and ESCRT-III component CHMP4C, then facilitates abscission exit. Generally, this manuscript is of interest in the cell cycle field and is helpful for people to understand the cell division more precisely. However, some evidences the authors presented are not convincing. These problems summarized below should be clearly solved before consideration for acceptance of this manuscript.

Major problems:

1. No rescue experiments. Since the authors have claimed that in yeast glutamic acid could potentially phospho-mimetic Aurora B phosphorylation at S227 in eukaryotes, they should carry this rescue experiment in DLD1 or HeLa cells to see whether the cells expressing Aurora B S227E (or S227D) mutant proteins separate normally during cytokinesis.

- S227E appears to be a poor phosphomimetic as approximately 50% of cells fail cytokinesis (see below). Also, when considering the conditional requirement for this phosphorylation, it may be hypothesized that constitutive phosphorylation of this site (in complex multicellular organisms) may not simply 'rescue' the phenotype we describe, instead removing the ability to regulate the timing of exit of cytokinesis, causing the cell to attempt abscission in the presence of DNA bridges.

They also need to do the rescue experiments with Borealin and CHMP4C

- we were unable to stably express GFP-Borealin S165E mutant in the DLD1 model used though-out the manuscript.
- Capalbo *et al* (2012) have already reported that expression of CHMP4C S210E is either a poor phosphomimetic or that CHMP4C phosphorylation/dephosphorylation cycles must be tightly regulated for proper function of the protein.

2. The authors use Proximity Ligation Assay to prove that PKC ϵ and Aurora B interacted in midbody(Fig1D), however, considering the limited space of midbody and normally very few PKC ϵ localizes in midbody(Figure S1A,S1B), this experiment is not very convincing without negative controls. It might be better for the authors to find out the interaction domain of PKC ϵ with Aurora B, and use the domain-depletion construct to perform this experiment.

- Experiments have been repeated using antibodies against the endogenous proteins (Figure 1D) and the negative controls are now included in the supplementary materials (S1D)
- Defining the domain of interaction for PKC ϵ would indeed be valuable for identifying its binding partner/s at the midbody but this is an extensive body of work we believe to be outside the scope this paper

3. Similar with the question above, the timing of Aurora B S227 phosphorylation by PKC ϵ should be carefully checked in the whole cell cycle. The S227 fluorescence staining and WB analysis with cells in different cell cycle stages are necessary. Moreover, the model (Fig. 4D) proposes that upon bridge resolution, PKC ϵ phosphorylates Aurora B, the authors should give more direct evidence to prove this conclusion.

- Figure S1F demonstrates localization of pS227 species of Aurora B through the preceding stages of mitosis – metaphase, anaphase and telophase, this is described in the text and commented on in the discussion. It was only possible to enrich cells in each phase of the cell cycle/mitosis, rather than synchronize cells which confounded western blot analysis.
- We feel that, when taken together, the data we have presented in this manuscript do support the working model proposed. We demonstrate that Aurora B Ser227 phosphorylation is lost upon PKC ϵ inhibition *in vitro* and *in vivo*. Throughout the manuscript we use expression of the Aurora B S227A mutant in parallel to PKC ϵ inhibition and consistently show that cells which attempt to complete cytokinesis with chromosome bridges, ultimately fail when this site is not or cannot be phosphorylated.

4. The authors claim that double phosphorylated Aurora B has greater kinase activity to Borealin. However, this experiment procedure description is too simple, making the result of the peptide array to be difficult to be understood (Fig.3A). The authors should make it clear that how double-phosphorylated Aurora B WT and single-phosphorylated S227A recombinant proteins were prepared and whether they were active.

- Information regarding testing of the recombinant proteins has been added to the methods: Recombinant Aurora B WT and S227A proteins were assessed

for their phosphorylation state (pS227 and pT232) and kinase activity against the substrate Histone H3 S10 prior to use in all assays. This data was included already in Figure S3. Further evidence that both preparations were active kinases is provided by the fact that the vast majority of substrates were able to be phosphorylated to a similar extent in the peptide array.

- No additional treatment was required of the WT protein for phosphorylation of S227. The preparations of protein were highly concentrated suggesting that under these conditions in bacteria Aurora B is able to accumulate phosphate at this site through autophosphorylation. It is noteworthy that when the protein is produced less efficiently (lower concentrations) the 227 site is not occupied.

5. The logical relationship between Fig.3 and Fig.4 is not strong. The functions of Borealin S165 phosphorylation and Aurora B S227 phosphorylation toward CHMP4C needs more direct evidence.

We have modified the text to provide the logical progression we consider exists. The key point being that while we have mapped a series of events that phenocopy each other providing a logic for their functional inter-connection, we do not demonstrate that this is connected to what is established as the AuroraB abscission checkpoint – it would be a reasonable conclusion but it is not one we have formally tested at this point. So we are exploiting the prior art that has defined CHMP4C knock-down as a means of by-passing the Aurora B checkpoint, to demonstrate that this knock-down ALSO by-passes the abscission failure of AuroraBS227A expressing cells. We feel this is crucial to tying up this issue.

Minor problems:

1. Fig.1C: no loading control of PKC ϵ . ATP should be labeled as 32P-ATP.
 - 80ng of kinases was loaded per lane, the methods and figure legend has been amended to reflect this. Labeling has been corrected.
2. Fig.S2A: no loading control of each sample and it's hard to understand why endogenous Aurora B has two lanes. Furthermore, GFP-Aurora B seems at least 5 times more than endogenous level, not just near the level as authors claimed.
 - representative western blots and loading controls have been included
3. FigS2B, Fig.4B and 4C: the cells counted in each experiment should be shown.
 - the text in each figure legend has been revised to reflect this.
4. FigS2E: the time of each still image should be shown.
 - The time of each frame has now been included
5. Fig.2D and Fig.3C: Only two independent experiments were carried out, how could the authors get the significant difference?
 - Figure 2D – this is a typographical error and has now been amended. We apologise for the confusion.
 - Figure 3C - This has been repeated in the new stable, inducible GFP-Borealin cell lines and the change has been reflected in the results, discussion and methodology.
6. Fig. S3A and S3B: The total loading amount of Aurora B is inconsistent. It seems

S277A is much more than WT, thus the conclusion that the two proteins possess equal activities is not convincing.

- The western blot in figure 3A has been repeated.
- The quantification of western blots in figure 3B has been normalized to the amount of Aurora B present in each assay and therefore represents the specific activity of Aurora B against Histone H3 Ser10. The axis of the graph has been re-labeled to reflect this and the figure legend has been revised.

Reviewer #3 (Remarks to the Author):

A. Summary of the key results

PKC ϵ has previously been found to have roles important to resolution of concatenated chromosomes as cells exit mitosis and to the function of ZO-1 and RhoA during cytokinesis. Here, further insight into PKC ϵ function is obtained through investigation of its regulatory interface with the Aurora B kinase.

To find substrates of PKC ϵ when it localizes to the intercellular bridge, a peptide array biased toward midbody proteins with potential recognition sites for phosphorylation was employed. One hit from this screen was Aurora B S227, a conserved residue near the auto-phosphorylation site T232. An S227 phospho-specific antibody was raised and used to confirm that PKC ϵ phosphorylates Aurora B. Results suggest that phosphorylation of S227 does not affect the phospho-status of T232. Despite this, expression of Aurora B S227A resulted in an increase in binucleated cells. This was confirmed by live-imaging experiments showing that expression of Aurora B S227A resulted largely in failed cytokinesis, similar to treatment with BLU577, a PKC ϵ inhibitor. In many cases, cells expressing Aurora B S227A had visible chromatin bridges. Yet, the timing of the midbody-stage did not seem to change under these conditions (authors should clarify, this was measured in the minority of cases where

midbodies were resolved -or does it include the timing of midbody regression).

The time taken to complete cell division was defined as the time at telophase onset (bundling of microtubules at the midbody, 'dumbbell' morphology) until complete breakdown of the Aurora B/tubulin positive midbody. The timing of cytokinesis/failure of cytokinesis in this assay is comparable to what Carlton *et. al.* (2012) report: abscission times of 116 ± 45 and 137 ± 61 minutes in control HeLa cells expressing GFP. We report here (in the GFP-Aurora B/mCherry-tubulin DLD1 cells) abscission times of 155.2 ± 53.57 minutes for WT expressing cells and 173.3 ± 54.24 minutes for S227A expressing cells.

Finding that intracellular targeting, catalytic activity, and ATP binding of Aurora B are not altered by phosphorylation of S227, the authors pursue the hypothesis that substrate specificity is modulated by this phospho-modification. To test this, a second peptide array is used (biased toward Aurora B substrate sites and comparing S227A to Aurora B capable of double phosphorylation). One of several positive (differential) hits in the above screen is tested functionally: site S165 in the CPC subunit Borealin. Expression of this mutant phenocopies the high rate of abscission failure observed when Aurora B S227A is expressed or when PKC ϵ is inhibited. The authors conclude that PKC ϵ is required to phosphorylate Aurora B when there is DNA retained in the

cleavage furrow, which in turn must phosphorylate Borealin at 165 in order to trigger completion of abscission -or to trigger an abscission checkpoint, allowing time for resolution of lagging DNA (but see critique below).

Closer inspection of phosphorylated Aurora B led to the appreciation that when phosphorylated at S227, Aurora B is exclusively at the central midbody ring. This localization pattern suggested a role in modulation of the abscission machinery, which is thought to be regulated at this site. Looking specifically at CHMP4C, the authors find that treatment with BLU577 or expression of Aurora B S227A alters CHMP4C targeting at the midbody (this is clear in the examples in Fig 4A, but not as easily seen in supplemental Fig 4).

Although the logic does not seem entirely clear-cut, the authors next test whether knockdown of CHMP4C would rescue abscission defects seen when Aurora B S227A is expressed. When the ability to execute an abscission checkpoint is abrogated by depletion of CHMP4C, cells no longer fail as frequently in abscission following Aurora B S227A expression. At face value, this means that abscission failure that results from lack of PKC ϵ signaling via Aurora B modification is due to stimulation of an abscission checkpoint coupled with an inability to progress forward from there or to sustain the checkpoint-arrested state.

B. Originality and interest:

The results presented are original and will be of high interest to the field.

C. Data & methodology:

One main question is about the proteins used to compare Aurora B to its S227A counterpart. While the relevant consideration is the effect of phosphorylation at 227 on binding and activity, the Methods describes the use of recombinant protein without clarifying how S227 is phosphorylated. i.e., is there a pre-incubation with PKC ϵ ? If so, how quantitative is this phosphorylation? (if only a minor proportion is modified, this too makes the comparison difficult) How is PKC ϵ removed?

- It was found that addition of PKC ϵ to the recombinant Aurora B WT protein was unnecessary as the protein was phosphorylated on the S227 site after purification. Recombinant Aurora B was produced with high efficiency and we find that under these conditions, Aurora B can accumulate phosphate in the 227 site presumably through autophosphorylation. We should add that while expression of high levels of Aurora B WT in *E. coli* did produce a doubly phosphorylated species this was not the case in less concentrated preparations. Also, we provide evidence that this autophosphorylation does not occur in vivo as when cells are treated with the PKC ϵ inhibitor, phosphorylation at the S227 site is absent while the established autophosphorylation site T232 is not PKC ϵ inhibitor sensitive.

One important conclusion is the change in substrate specificity of Aurora B when phosphorylated at S227. Specific aspects of this conclusion need to be more rigorously addressed. First, clarify how many times the peptide array was performed.

- The peptide array was performed as a tool to screen substrates of Aurora B for the purposes of subsequent validation if found to be preferentially phosphorylated by either the 1P or 2P form of the kinase, with a view to take forward candidate 'hits' for further validation and assessment. We demonstrate for the substrate protein Borealin that the differential is clearly observed (see also below) and specifically contrasts with equivalent activity towards histone H3. Note the latter was confirmed as being equally phosphorylated by both forms in the peptide array as found in solution assays.

Second, test changes in activity towards substrates using in vitro kinase assays.

- A kinase assay using Borealin S165 phosphorylation as the substrate has now been included in Figure 3B demonstrating good activity towards this site for the doubly phosphorylated Aurora B but not the S227A mutant.

Additional experiments are required to convincingly demonstrate a switch in specificity for Aurora B and its dependence on PKC ϵ . The data show that phosphorylation of S165 in Borealin is important, but whether this strictly depends on PKC ϵ is not conclusive.

- We raised phosphospecific antibodies against the S165 site of Borealin but these were unable to be used for IF. We did however show in an in vitro kinase assay that recombinant Aurora B WT protein was able to phosphorylate this site while the S227A mutant could not.
- We have demonstrated consistently throughout this manuscript that in every situation Aurora B S227A expression phenocopies PKC ϵ inhibition. Based on this, in the experiments demonstrating phosphorylation of Borealin S165 by Aurora B, we use the S227A mutant as a surrogate for PKC ϵ inhibition. Taken together with the in vitro data, the fact that PKC ϵ inhibition, Aurora B S227A and Borealin S165A all phenocopy provides evidence that these events sit on the same pathway, initiated by PKC ϵ .

Other points:

Figure S1: indicate which is Aurora B on the heat-map; what does the arrowhead indicate?

- The arrowhead has been removed, Aurora B pS227 is position B1 and this has been indicated by a green box.
- Include more controls for the PLA (i.e., a negative control for background signal)
- Negative control for PLA has now been included in Figure S1D
- Explain the M486A mutation used
- For clarity, experiments using the PKC ϵ M486A mutation have been removed from the manuscript.
- Is the BIM inhibitor the same as BLU577/compound 18? If not, why are different inhibitors used in vitro and in vivo?
- BIM is an inhibitor of both classical and novel PKC isoforms, whilst BLU577/compound 18 is a more selective inhibitor of the novel PKC isoforms. Similar results have been obtained *in vivo* with BIM as described in the manuscript but we felt it prudent to include the more PKC ϵ selective inhibitor.

-Part E, This signal is not very compelling and is based on single images (presumably the best). Is there a way to quantify?

- Images have been replaced with higher resolution black and white images to demonstrate S227 phosphorylation at the midbody in Figure 1E, S1E, S1F Referring to the site "being occupied" is confusing (top of third page).
- The text has been amended for clarity here.

Figure S4C -the classification of early and late cytokinesis is not clear, nor is the "arm" vs. midbody ring in the images. This needs to be presented more clearly (use arrows) and in a more quantitative manner.

- Images have been labeled to define these structures.

D. Appropriate use of statistics and treatment of uncertainties

See above for comments that have to do with quantification and reproducibility.

- Figures and text have been amended and described as indicated

E. Conclusions: robustness, validity, reliability

Several novel findings are presented in this manuscript, but there are certain results that need to be more rigorously established.

- We feel the additional data and improved image quality provides the robust dataset to support the conclusions drawn/working model

F. Suggested improvements: experiments, data for possible revision

In addition to addressing the points brought up above (such as quantitative assessment of change in phosphorylation specificity for Aurora B), one aspect of this manuscript that was challenging was the lack of integration with previous findings on the role of PKC ϵ . For instance, the high rate of Lap2 β positive bridges when Aurora B S227A is expressed seems likely to reflect a role of this signaling pathway in preventing concatenation, consistent with previous reports for PKC ϵ function. This same pathway appears to be important to resolving the bridge in coordination with an abscission checkpoint.

- This has been examined further in the discussion.

When explaining Borealin as a downstream target of PKC ϵ -Aurora B signaling, it was suggested that cells could not respond to DNA in the cleavage furrow when the phospho-site in Borealin was mutated --but, it was unclear why so many cells would have DNA in the cleavage furrow to begin with (87% fail in cytokinesis).

- Our previous work (Brownlow et al 2014) demonstrated that DLD1 cells have a defective G2 catenation checkpoint, meaning that they are more likely to enter mitosis with residual catenation resulting in an increase in the number of chromatin bridges as they traverse metaphase and anaphase. We see evidence of this in the Aurora B S227A mutant cells in the presence of PICH positive structures in anaphase, increasing the likelihood of residual DNA in the cytokinetic furrow. This of course does not preclude a role for phosphorylation of this S227 site earlier in mitosis as well as cytokinesis. We examine this idea further in the discussion.

Finally, to test the working model that CHMP4C depletion allows cells to progress forward despite chromatin bridges (which, again, are elevated when PKC ϵ is inhibited), downstream events such as DNA damage and chromosomal instability would be predicted to be prevalent and the manuscript would be strengthened by their assessment.

- Carlton *et. Al* (2012) have previously reported that long term (3 weeks) knockdown of CHMP4C with shRNA leads to an accumulation of cells with H2AX (pS139) and 53BP1 foci. However in our experiments (24h siRNA) we see no evidence of DNA damage caused by loss of CHMP4C (see below). We do see an increase by immunoblotting of H2AX (pS139) in response to 24h BLU577 treatment concomitant with an increase in the presence of micronuclei in these cells which may be attributed to a prior mitotic mis-segregation event. While these observations are interesting and speak to an earlier role for PKC ϵ and Aurora B S227 phosphorylation in mitosis, it is outside the scope of the current manuscript but an interesting possibility.

G. References: appropriate credit to previous work?

Seemed appropriate

H. Clarity and context

See problems discussed with integrating previous results and making the logic of the CHMP4C experiment more clear. A different discussion point that should be mentioned is the limitation of using known phospho-sites to screen for altered specificity of Aurora B.

- The results and discussion have been updated to address the issues regarding the link between Aurora B – Borealin – CHMP4C and we hope this is now clear and concise.
- We have included a discussion point regarding the limitations of the putative phosphosites screened in the context of the assays we describe.

Reviewers' Comments:

Reviewer #1 (Remarks to the Author)

I have looked carefully at the revised version of the manuscript and the authors' response to my comments. This is a much improved version of the paper and my suggestions have been addressed whenever possible. I believe the manuscript is now in a much better shape and the additional data presented further support the authors' conclusions. I have no further criticisms or comments and believe that the work will be of significant interest to the readership of Nature Communications.

Reviewer #2 (Remarks to the Author)

This revised version of the manuscript NCOMMS-16-00122A and the point-by-point response have carefully dealt with my concerns. Now I have no further questions to this revised manuscript, and therefore recommend it to be published in the prestigious journal Nature Communications.

Reviewer #3 (Remarks to the Author)

The authors have addressed several points brought up in review and the manuscript contains a description of a connection between PKC ϵ , Aurora B, and Borealin that is novel and interesting. Yet, it remains difficult to conclude that this pathway operates to control the resolution of abscission: given the earlier roles for PKC ϵ in mitosis, it seems equally possible that the pathway leads to increased DNA bridges and activation of the abscission checkpoint. The observation that more LAP2 bridges are detected when cells express AurB S227A suggests this is the case (vs. the same number detected, but more readily resolved when wild-type AurB expressed). In addition to this underlying conceptual problem, there are issues that remain with the data. Perhaps most significant, while improved images were added, many experiments still rely on an image or two to support the conclusion. Quantification is required to demonstrate how the localization of various proteins changes with phosphorylation state etc. This means both 1) an objective scoring method (or a subjective method applied blindly) and 2) scoring a number of events such that the change can be evaluated for statistical significance. In addition, although Figure 3B, an in vitro kinase assay of Borealin, is included to bolster the documentation for altered specificity of Aurora B following S227 phosphorylation, this blot has spurious bands and is very difficult to interpret.

A couple additional specific points:

While the authors describe in review comments the importance of Aurora B concentration in order to obtain recombinant Aurora B phosphorylated at S227, I didn't see this in the manuscript. The information is key to others repeating the experiment.

I still did not see information on how early and late cytokinesis were defined.

The authors add analysis of different cell cycle stages with the AurB p227 antibody, but this is not meaningful since they have also demonstrated that the bulk of signal detected with this antibody is non-specific. What happens to the mitotic signal under knockdown conditions?

I think it should be more clearly stated in the text/legend that timing to complete cell division includes both successful and failed cytokinesis.

Reviewers' comments:

Reviewer #1 (Remarks to the Author):

I have looked carefully at the revised version of the manuscript and the authors' response to my comments. This is a much improved version of the paper and my suggestions have been addressed whenever possible. I believe the manuscript is now in a much better shape and the additional data presented further support the authors' conclusions. I have no further criticisms or comments and believe that the work will be of significant interest to the readership of Nature Communications.

Reviewer #2 (Remarks to the Author):

This revised version of the manuscript NCOMMS-16-00122A and the point-by-point response have carefully dealt with my concerns. Now I have no further questions to this revised manuscript, and therefore recommend it to be published in the prestigious journal Nature Communications.

Reviewer #3 (Remarks to the Author):

The authors have addressed several points brought up in review and the manuscript contains a description of a connection between PKC ϵ , Aurora B, and Borealin that is novel and interesting. Yet, it remains difficult to conclude that this pathway operates to control the resolution of abscission: given the earlier roles for PKC ϵ in mitosis, it seems equally possible that the pathway leads to increased DNA bridges and activation of the abscission checkpoint. The observation that more LAP2 bridges are detected when cells express AurB S227A suggests this is the case (vs. the same number detected, but more readily resolved when wild-type AurB expressed).

- While we do not exclude the possibility that PKC ϵ phosphorylation of Aurora B during mitosis may play a role in the resolution of chromosome segregation errors, we believe both our previously reported and current observations reflect a distinct role for PKC ϵ and Aurora B signalling during exit from the abscission checkpoint. Specifically, Saurin et al. (Saurin AT, et al. *Nature Cell Biology*, 10, 891-901(2008); Saurin AT, et al. *Cell Cycle*, 8, 549-555(2009)) demonstrate that inhibition of PKC ϵ during cytokinesis in HEK293 cells results in an accumulation of the kinase at the midbody and that inhibition results in an increased frequency of cells which fail cytokinesis. We also observe a loss of Aurora B pS227 immunoreactivity within 30 minutes of PKC ϵ inhibition (Figure 1E and Supplementary Figure 1F) suggesting that the latest mitotic phase these cells could be in when PKC ϵ is inhibited would be anaphase (this was perhaps unclear in the previous version and we have updated the main text to emphasise this experimental detail). Taken together, these

findings indicate that accumulation of inhibited PKC ϵ at the midbody during cytokinesis delays exit and this may be attributed to the inability to phosphorylate Aurora B at S227 to signal for exit from the abscission checkpoint.

- We note the significant difference in the number of Lap2B positive bridges observed in Figure 2D (WT 6.25% \pm 12.5 vs S227A 43.125% \pm 19.2) and apologise if this was unclear in previous versions of the manuscript. We did not comment on the resolution of these bridges but the number of Lap2B positive bridges closely correlates with the number of cells which go on to fail cytokinesis and become binucleate cells (Binucleate cells: WT 6.25% \pm 12.5 vs. S227A 80.9% \pm 21.5 for this dataset)

In addition to this underlying conceptual problem, there are issues that remain with the data. Perhaps most significant, while improved images were added, many experiments still rely on an image or two to support the conclusion. Quantification is required to demonstrate how the localization of various proteins changes with phosphorylation state etc. This means both 1) an objective scoring method (or a subjective method applied blindly) and 2) scoring a number of events such that the change can be evaluated for statistical significance.

- The key observations of Aurora B S227 phosphorylation and CHMP4C localisation have been quantified. Scored images describe: 1) changes in phospho-specific signal (the presence or absence of Aurora B pSer227 signal at the midbody -Figure 1E) and 2) changes in the localisation (CHMP4C – Figure 4C and D, Supplementary Figure 5D).
- We have also included the following text in the figure legends for clarity
 - Figure 1E: A minimum of 30 high resolution, single cell images per condition from 12 experiments in 2 different cell lines were acquired, a representative image is shown here.
 - Figure 4C : A minimum of 12 high resolution, single cell images per condition from 4 experiments were acquired, a representative image is shown here.
 - Figure 4D: A minimum of 5 high resolution, single cell images per condition from 2 experiments were acquired, a representative image is shown here.
 - Supplementary Figure 5D: A minimum of 11 high resolution, single cell images per condition from 4 experiments were acquired, a representative image is shown here.

In addition, although Figure 3B, an in vitro kinase assay of Borealin, is included to bolster the documentation for altered specificity of Aurora B following S227 phosphorylation, this blot has spurious bands and is very difficult to interpret.

- We have included the full western blot in the main figure now. We note that while the phospho-specific sera used to probe the immunoblot does detect some non-specific bands, upon reprobing with the total Borealin

antibody we were able to demonstrate that the lower band that has been marked Borealin pS165 is in fact Borealin.

A couple additional specific points:

While the authors describe in review comments the importance of Aurora B concentration in order to obtain recombinant Aurora B phosphorylated at S227, I didn't see this in the manuscript. The information is key to others repeating the experiment.

- This information has now been included in the methods for reproduction of the experiment.

I still did not see information on how early and late cytokinesis were defined.

- Definition of early and late cytokinesis is included in the figure legend of Supplementary figure 5B as well as the main text as follows: During early cytokinesis, CHMP4C is present on the midbody arms, adjacent to the midbody ring (as can be demonstrated by two peaks in the green channel on the pixel intensity profiles), whilst during late cytokinesis it is concentrated within the midbody ring (single green peak of the pixel intensity profile)(Supplementary Figure 5B).

The authors add analysis of different cell cycle stages with the AurB p227 antibody, but this is not meaningful since they have also demonstrated that the bulk of signal detected with this antibody is non-specific. What happens to the mitotic signal under knockdown conditions?

- This was included in Supplementary Figure 1E (bottom panel). The following text has been included for clarity: Analysis of S227 phosphorylation through mitosis revealed chromatin associated staining of Aurora B S227 phosphorylation in mitosis (which was absent following Aurora B knockdown) but not telophase.

I think it should be more clearly stated in the text/legend that timing to complete cell division includes both successful and failed cytokinesis.

- We have now amended the text and figure legend to clearly state this.

Reviewers' Comments:

Reviewer #3 (Remarks to the Author)

The authors have addressed questions and concerns from the previous review -the manuscript has been strengthened and will be of high interest to the field.